# Aqueous pulsed electrochemistry promotes C−N bond formation via a one-pot cascade approach

Meng He[1], Yongmeng Wu [1]✉, Rui Li [1], Yuting Wang [1], Cuibo Liu [1] & Bin Zhang [1]✉

Electrocatalytic C−N bond formation from inorganic nitrogen wastes is an emerging sustainable method for synthesizing organic amines but is limited in reaction scope. Integrating heterogeneous and homogeneous catalysis for one-pot reactions to construct C−N bonds is highly desirable. Herein, we report an aqueous pulsed electrochemistry-mediated transformation of nitrite and arylboronic acids to arylamines with high yields. The overall process involves nitrite electroreduction to ammonia over a Cu nanocoral cathode and subsequent coupling of $NH_3$ with arylboronic acids catalyzed by in situ dissolved Cu(II) under a switched anodic potential. This pulsed protocol also promotes the migration of nucleophilic $ArB(OH)_3^-$ and causes the consumption of $OH^-$ near the cathode surface, accelerating C−N formation and suppressing phenol byproducts. Cu(II) can be recycled via facile electroplating. The wide substrate scope, ready synthesis of $^{15}N$-labelled arylamines, and methodological expansion to cycloaddition and Click reactions highlight the great promise.

The construction of C−N bonds has attracted much attention due to nitrogen-containing compounds being widely used in chemical and pharmaceutical industries as well as in materials science[1–4]. However, high-temperature and high-pressure reaction conditions cause serious environmental pollution problems[5]. Thus, it is important to explore facile synthetic strategies for C−N bond formation. Recently, the formation of electrochemically coupled C−N bonds driven by renewable energy sources has emerged as a promising sustainable strategy[6–8]. Coupling inorganic nitrogen sources (e.g., nitrogen ($N_2$), $NO_3^-$, $NO_2^-$) and carbon sources (e.g., carbon dioxide, carbon monoxide, and other carbonyl substrates) to form upgraded organic amines is an emerging hot field in electrocatalytic C−N bond construction[9–15]. For example, Wang et al. reported electrocatalytic C−N coupling to successfully synthesize methylamine from $CO_2$ and $NO_3^-$ catalyzed by a well-designed molecular cobalt catalyst[13]. Our group realized the electrosynthesis of formamide from formic acid with $NO_2^-$ over a nanostructured copper (Cu) cathode[4]. Despite these achievements, C−N bond construction mainly starts from small nitrogen and carbon sources through (i) condensation of hydroxylamine with carbonyl intermediates or highly active carbonyl substrates and (ii) cross-coupling of in situ formed nitrogen and carbon free radical intermediates. However, the current studies are limited in reaction scope, impeding methodology development and applications. Although electrocatalytic C−N coupling by using additional $NH_3$ has been reported[16,17], the electrosynthesis of organic amines from $NH_3$ generated in situ via $NO_3^-/NO_2^-$ electroreduction has rarely been studied owing to the relatively complex collection and use of $NH_3$ and lower reactivity over other nitrogen intermediates. Therefore, exploring a reaction mode to expand the scope of electrochemical C−N bond formation to obtain various amines and clarifying the underlying mechanism are urgently needed.

Arylamines are essential structural units and building blocks in many natural products and pharmaceuticals[18,19]. Transition-metal-catalyzed Chan-Lam coupling of $NH_3$ with arylboronic acids has been

[1]Department of Chemistry, School of Science, Institute of Molecular Plus, Tianjin University, Tianjin 300072, China. ✉e-mail: ymwu01@tju.edu.cn; bzhang@tju.edu.cn

developed as a mild and effective synthetic tool to access various arylamines[20,21]. However, the preparation of $NH_3$ that dominantly relies on the Haber-Bosch process is highly energy intensive with a large amount of $CO_2$ emissions. Additionally, the collection, transport, storage, and usage of $NH_3$ are time- and manpower-consuming processes that require high costs and complex handling and cause safety risks owing to its pungent property. Inspired by recent advances in electrochemical C − N coupling based on inorganic nitrogen sources[9–17], we propose to explore a highly promising and sustainable method by integrating $NO_3^-/NO_2^-$ electroreduction to $NH_3$ with cascade metal-catalyzed cross-coupling of $NH_3$ with arylboronic acids to form arylamines, which remain untouched up to now. One key issue is that homogeneous metal complexes for the C − N coupling step must be added, and most metal complexes are easily reduced to zero valence metals under cathodic conditions for $NO_3^-/NO_2^-$ reduction, leading to their deactivation for C − N coupling. Furthermore, nearly all metal electrodes will be oxidized and leach metal ions ($M^{\delta+}$) into the electrolyte under certain anodic potentials[22,23]. The dissolved $M^{\delta+}$ has been applied to the in situ preparation of self-supported metal−organic frameworks (MOFs) or metal complexes[24,25]. Therefore, the utilization of in situ dissolved $M^{\delta+}$ generated at an anodic potential for catalyzing C − N bond formation to merge cathodic $NH_3$ from $NO_3^-/NO_2^-$ electroreduction for tandem synthesis of arylamines is exceptionally significant but remains an unsolved challenge.

Herein, by selecting the classical Chan-Lam coupling as a model case, we report a pulsed potential technique that promotes the one-pot two-step cascade transformation of $NO_2^-$ and arylboronic acids to arylamines in an aqueous electrolyte. The method includes the electroreduction of $NO_2^-$ to $NH_3$ over a low-coordinated Cu nanocoral (LC-Cu NC) cathode and subsequent C − N coupling of $NH_3$ with arylboronic acids catalyzed by in situ generated Cu(II) via electrooxidation of the same Cu electrode (Fig. 1). Our pulsed method can produce Cu(II) via in situ electrooxidation of the Cu electrode rather than extra addition, which is favorable for lowering the cost and simplifying the operation procedure. Then, a pulsed electrochemical setup can also assist oxidative turnover and maintain high concentrations of Cu(II) without additional ligands since potentiostatic electrolysis induces irreversible copper plating, which offers a great opportunity

for achieving long-term electrolysis mediated by a homogeneous metal catalyst. In addition, pulsed electrochemistry can obviously suppress the formation of phenol byproducts by adjusting the pH of the solution. A series of ex and quasi-in situ techniques are adopted to comprehensively characterize the material with its transformation and key reaction intermediates to rationalize our speculations. This method can be not only applied to the efficient synthesis of functionalized primary arylamines and their labelled analogs but also expanded to both cycloaddition reactions and Click reactions for synthesizing 3,5-disubstituted isoxazoles and 1,2,3-triazole molecules.

## Results

### Consideration factors of electrode materials and pulsed electrochemical method

To achieve one-pot conversion of $NO_2^-$ and arylboronic acids to arylamines, tandem $NO_2^-$ electroreduction to $NH_3$ followed by C − N coupling of $NH_3$ with boronic acids should be well interconnected (Fig. 1). The key is to develop a highly active electrode and catalyst for each step. Cu-based materials are prevailing electrocatalysts for $NO_2^-$ reduction to $NH_3$ due to their low cost and good catalytic activity[26]. Tuning the electronic structure and/or coordinated surroundings can further enhance their electrochemical performances. We suppose that the low-coordinated Cu electrode, which can be readily produced via electroreduction of copper oxide or hydroxide precursor[27,28] with abundant active sites to facilitate the adsorption of $NO_2^-$ will be conducive to improving the selectivity and Faradic efficiency (FE) of $NH_3$ (Fig. 2b). The efficient production of $NH_3$ is highly significant to the next C − N coupling step with boronic acids. In addition, Cu catalysts are widely applied in Chan-Lam coupling reactions[20,21]. Therefore, Cu materials may be the ideal choice to implement the cascade transformation from $NO_2^-$ to valuable organic amines. Cu(II) is proven to be the active center for Chan-Lam coupling reactions[20,21]. Metallic Cu is easily oxidized under proper anodic potentials to leach Cu(II) into the electrolyte[29–31]. We propose to produce Cu(II) via in situ electrooxidation of the Cu electrode rather than extra addition, which is favorable to lower the cost and simplify the operations. Furthermore, phenol, one of the main byproducts for the coupling of ammonia/amines with boronic acids in aqueous solutions[20,21], would be promoted by

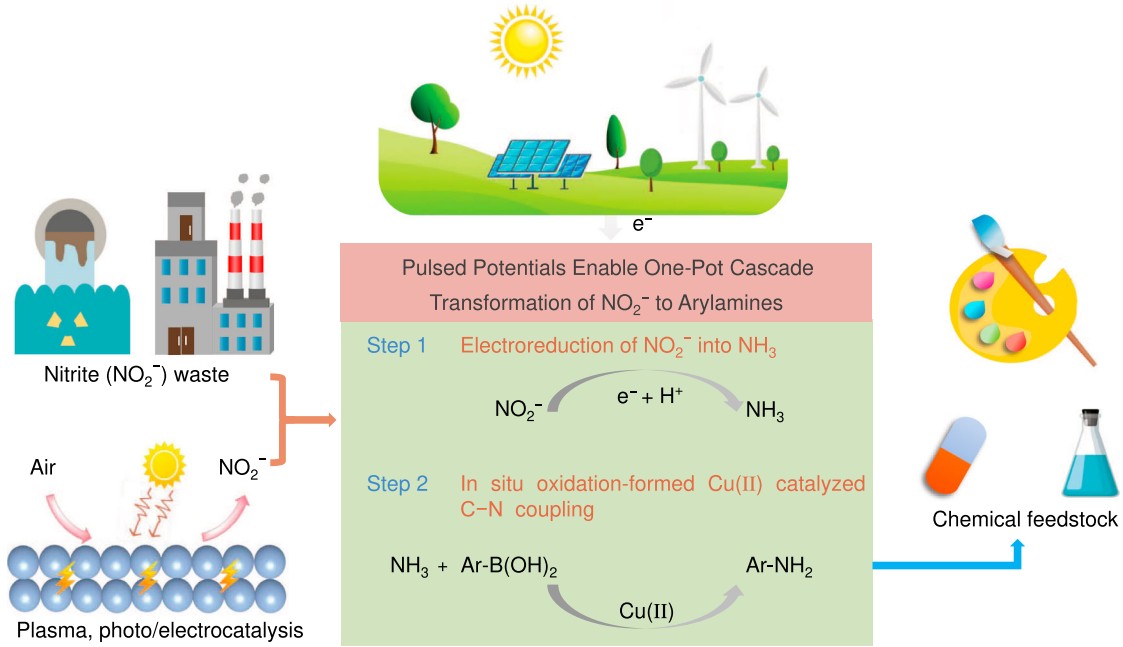

**Fig. 1 | The proposed pulsed electrochemical method.** Pulsed electrochemistry enables the one-pot transformation of $NO_2^-$ and arylboronic acids to arylamines via cascade reductive hydrogenation of $NO_2^-$ and in situ oxidation-formed Cu(II)-catalyzed C − N coupling.

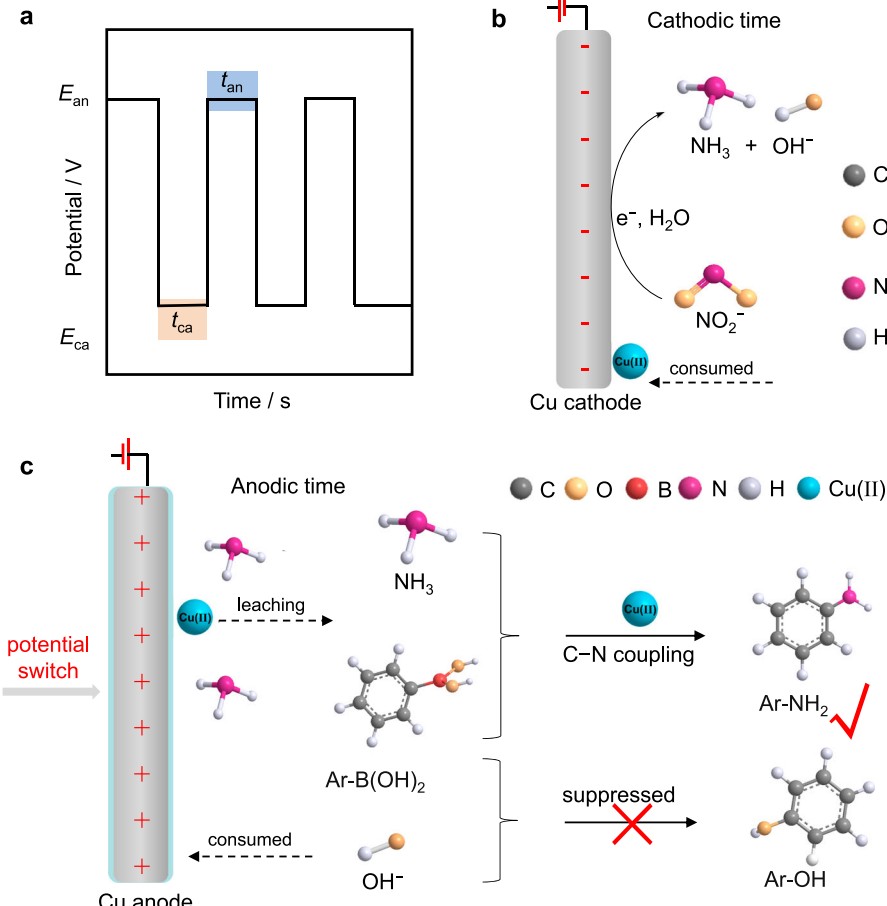

**Fig. 2 | The consideration factors for electrode materials and the proposed pulsed process. a** Pulsed potential waveform, **b** $NO_2^-$ reduction before pulsed electrolysis, **c** Cu(II) generation and Chan-Lam coupling under the anodic potential ($E_{an}$).

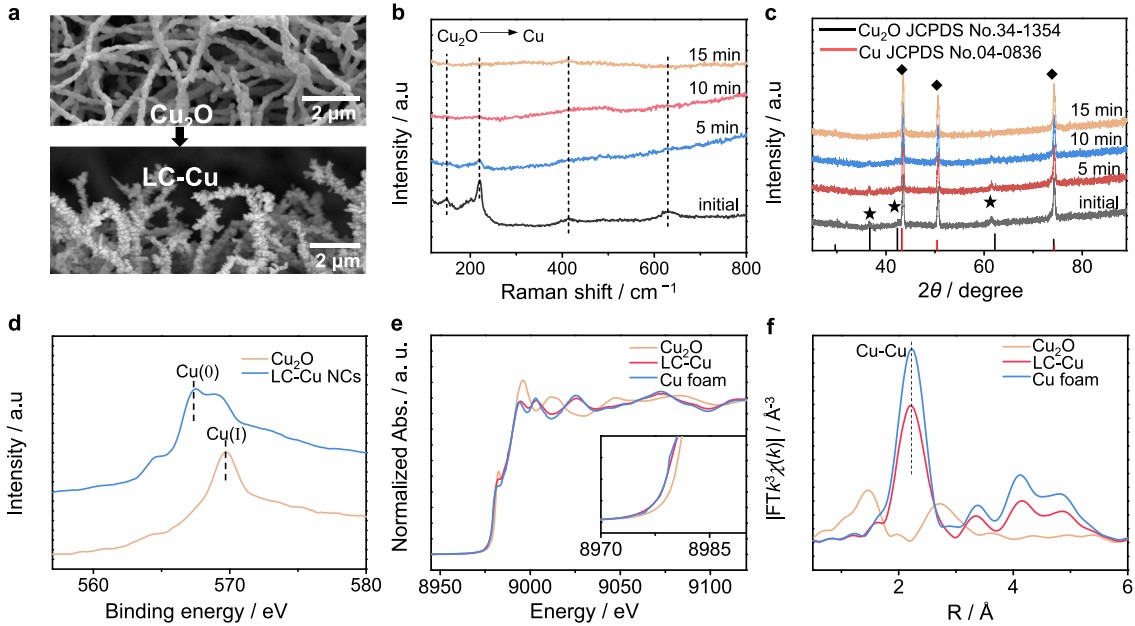

**Fig. 3 | Synthesis and characterization of low-coordinated Cu nanocorals.**
**a** SEM images of $Cu_2O$ and LC-Cu NCs. **b** In situ Raman spectra collected at −0.8 V vs. Hg/HgO in 0.25 M PBS electrolyte. **c** In situ XRD patterns and **d** Cu LMM AES spectra of $Cu_2O$ and Cu NCs. **e** Cu K-edge XANES spectra and **f** EXAFS spectra of Cu foam, $Cu_2O$, and LC-Cu NCs.

increasing the concentration of OH⁻ ions (Supplementary Fig. 1 and Supplementary Note 1)[32,33]. Although the pH value increases during $NO_2^-$ electroreduction, we think that reverse electrooxidation will consume OH⁻ ions due to electrode self-oxidation or the oxygen evolution reaction (OER)[29,30,34,35], hence suppressing undesired phenol byproducts (Fig. 2c). However, continuous electrooxidation is unfavorable owing to the need for a catalytic amount of Cu(II) for the Chan-Lam coupling reactions and the easy oxidation nature of $NH_3$/arylamines[36,37]. Third, pulsed electrochemical methods (e.g., square potential waveforms) can induce alternate oxidation and reduction by controlling pulsed frequencies (Fig. 2a), which will alter the concentration of substrates or intermediates in the electrical double layer (EDL) to affect the product distributions[38–45]. Based on these considerations, we propose that the pulsed technique will provide an ideal platform to realize the cascade conversion of $NO_2^-$ to organic amines through sequential $NO_2^-$ electroreduction over a Cu cathode and C − N coupling catalyzed by in situ dissolved Cu(II) by anode self-oxidation (Fig. 2a–c).

## Synthesis and characterization of low-coordinated Cu nanocorals

We synthesized self-supported $Cu_2O$ nanocorals (NCs) precursors on Cu foam via the thermal treatment of $Cu(OH)_2$ nanowires (NWAs) under an Ar atmosphere (Supplementary Fig. 2 and Supplementary Note 2). The scanning electron microscopy (SEM) images, X-ray diffraction (XRD) pattern, and Cu LMM AES spectra suggest the successful preparation of $Cu_2O$ NCs (Fig. 3a, c, and d). Then, LC-Cu NCs are fabricated through facile electroreduction of $Cu_2O$ NCs in 0.25 M phosphate-buffered saline (PBS, pH = 5.8) at −0.8 V vs. Hg/HgO (Supplementary Fig. 3 and Supplementary Note 3) (potentials in this work are all referred to as Hg/HgO unless stated otherwise). SEM images reveal the maintenance of the nanocoral morphology after electroreduction (Fig. 3a). However, the surface becomes rough, exposing more active sites, which can promote the adsorption of substrates and thus enhance the reaction efficiency. In situ Raman spectra show that the characteristic peaks of $Cu_2O$ at approximately 148, 220, 420, and 625 cm⁻¹ vanish within 15 min (Fig. 3b), suggesting the fast conversion of $Cu_2O$ to Cu. In situ XRD is also carried out to monitor the phase transformation of $Cu_2O$. As depicted in Fig. 3c, after electrolysis for 15 min, all the diffraction peaks in the XRD pattern are indexed to the pure Cu phase (JCPDS no. 04-0836), corresponding to the in situ Raman results. The Cu LMM AES spectrum was employed to distinguish Cu(I) and Cu(0) (Fig. 3d), and the peaks located at 567.5 eV belong to Cu(0)[26–27], which shift slightly toward low energy regions compared with those of Cu(I).

Furthermore, X-ray absorption spectroscopy (XAS) is used to determine the electronic structure and coordination environment of LC-Cu NCs. As shown in Fig. 3e, the Cu K-edge X-ray absorption near-edge structure (XANES) of LC-Cu exhibits similar features to that of the Cu foam, revealing that the reduced sample mainly consists of metallic Cu. In addition, the Fourier transformed $k^3$-weighted Cu-K edge EXAFS spectra show the new appearance of the Cu−Cu path in the reduced sample (approximately 2.23 Å), while the average Cu−Cu coordination number of LC-Cu (9.75) is significantly lower than that of Cu foam (12) (Fig. 3f, Supplementary Fig. 4, Supplementary Table 1). These results demonstrate that LC-Cu NCs formed via electroreduction of $Cu_2O$ NCs possess more low-coordination sites, which can be expected to facilitate electroreduction of $NO_2^-$ with high performance.

## One-pot C − N coupling by integrating heterogeneous and homogeneous catalysis

The electrolyte solvent and pH play an important role in reaction selectivity due to phenol byproducts being easy to form at a high concentration of OH⁻ ions in the Chan-Lam coupling reaction[32,33]. In addition, the efficient production of $NH_3$ from $NO_2^-$ electroreduction is highly significant to the next C − N coupling step. $NO_2^-$ electroreduction can proceed over the entire pH range, but the competing hydrogen evolution reaction (HER) lowers the ammonia selectivity and Faradaic efficiency (FE). The HER is serious in an acidic electrolyte, and phenylboronic acid is not stable in alkaline media[46,47]. Thus, the plused experiments are performed in a nearly neutral buffer. Furthermore, the hydrogen bonding interaction between MeOH and $H_2O$ inhibits the $H_2O$

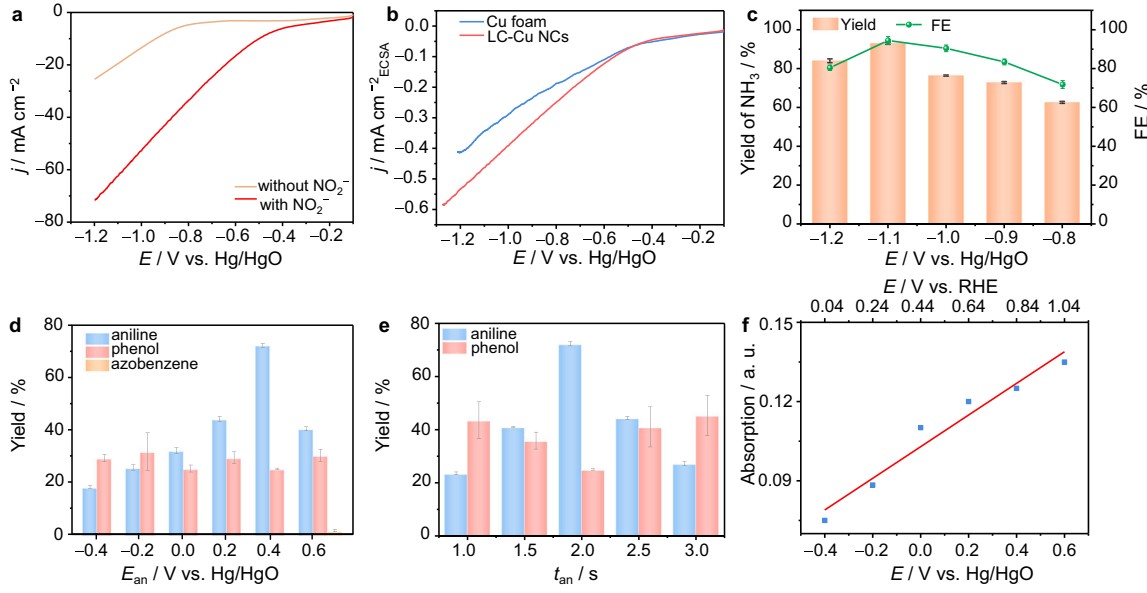

**Fig. 4 | Electrochemical performance. a** LSV curves of LC-Cu NCs at a scan rate of 5 mV s⁻¹ in a mixed solution of 0.25 M PBS and MeOH (2:1 *v/v*) with and without 2.0 mmol of $NO_2^-$. **b** LSV curves of Cu NCs and Cu foam at a scan rate of 5 mV s⁻¹ in a mixed solution of 0.25 M PBS and MeOH (2:1 *v/v*) with 2.0 mmol of $NO_2^-$. **c** Potential-dependent yields and FEs of $NH_3$ via $NO_2^-$ reduction over LC-Cu in the mixed 0.25 M PBS and MeOH (2:1 *v/v*) solution. **d** Isolated yields of the products under pulsed electrolysis conditions with $E_{ca}$ = −1.1 V, different $E_{an}$ values, and $t_{an}$ = $t_{ca}$ = 2 s. **e** Isolated yields of the products under pulsed electrolysis conditions: $E_{ca}$ = −1.1 V, $t_{ca}$ = 2 s; $E_{an}$ = 0.4 V, the different $t_{an}$ values. **f** Concentration-absorbance calibration curves of Cu(II) under pulsed electrolysis conditions with $E_{ca}$ = −1.1 V, different $E_{an}$ values, and $t_{an}$ = $t_{ca}$ = 2 s for 20 min.

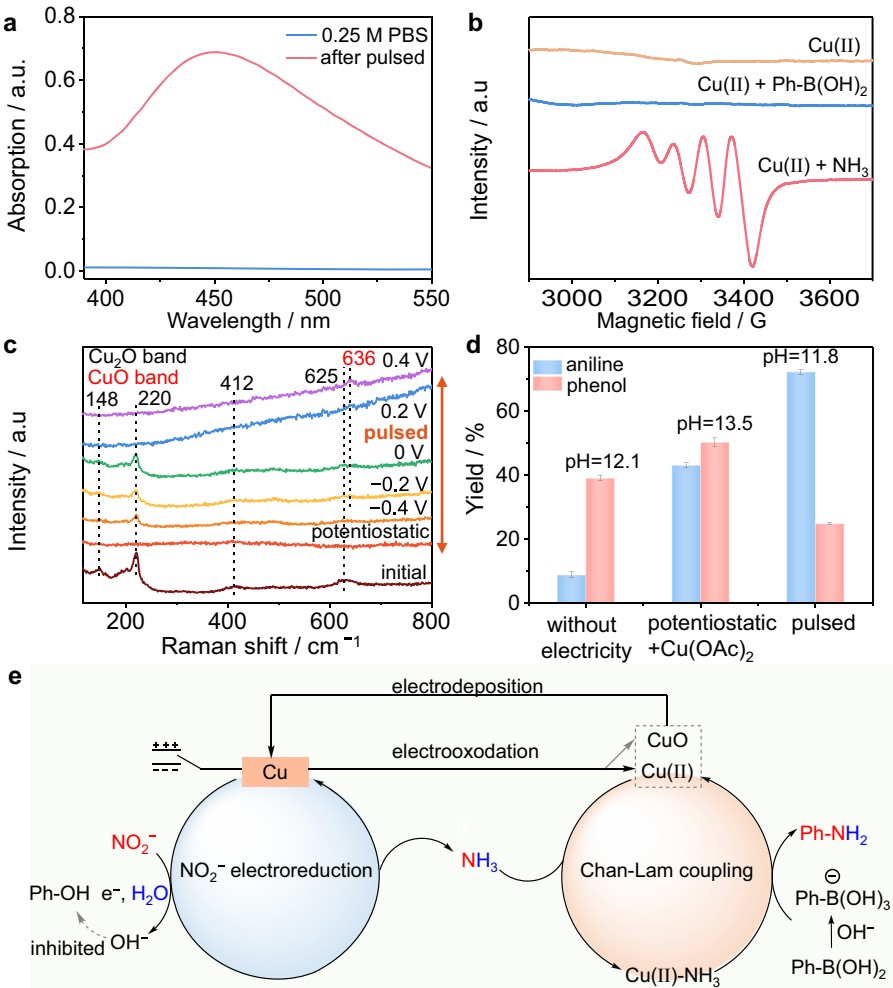

**Fig. 5 | Mechanism study. a** UV–vis spectra for the detection of Cu(II) after pulsed electrolysis. **b** X-band EPR spectrum of the Cu(II)·NH$_3$ complex during the reaction. Conditions: Cu(II)·NH$_3$ taken from the solution after pulse electrolysis. **c** In situ Raman spectra collected under potentiostatic measurement at −1.1 V and pulsed electrolysis conditions with different $E_{an}$ values in a mixed solution of 0.25 M PBS and MeOH (2:1 $v/v$). **d** Comparisons of the one-pot NO$_2^-$ to 2a under different conditions, including without electricity, pulsed conditions, and potentiostatic conditions with additional Cu(II) after potentiostatic conditions for 7 h to reduce NO$_2^-$ to NH$_3$. **e** A proposed mechanism.

dissociation process (the Volmer step) and slows the rate of migration of H$_2$O to the catalyst surface as hydrogen bonds[48], thus suppressing the HER and boosting the NH$_3$ FE from NO$_3^-$/NO$_2^-$ electroreduction. With this in mind, a mixture of MeOH and phosphate buffered saline (PBS) is adopted as the electrolyte for one-pot C−N coupling. We examined the performance of NO$_2^-$ electroreduction over the LC-Cu cathode. The linear sweep voltammetry (LSV) polarization curve displays a sharp increase in current density after adding 2.0 mmol of NaNO$_2$ into the anodic solution (Fig. 4a), indicating a much easier reduction of NO$_2^-$ than water. After normalization by the electrochemical surface area (ECSA; Supplementary Fig. 5), LC-Cu NCs still require a smaller overpotential to achieve a benchmark current density (Fig. 4b). Meanwhile, NH$_3$ is generated at a higher TOF and yield rate over LC-Cu NCs than the Cu foam (Supplementary Fig. 6 and Supplementary Note 4), implying the high activity of LC-Cu NCs toward NO$_2^-$ electroreduction.

Next, we examine the feasibility of a one-pot transformation of NO$_2^-$ and arylboronic acids to arylamines by using the pulsed electrochemical method that periodically applies an anodic potential ($E_{an}$) and a cathodic potential ($E_{ca}$). Figure 4c reveals that −1.1 V is optimal for NO$_2^-$ electroreduction, giving rise to NH$_3$ with the highest yield and 95% FE over LC-Cu NCs. We thus select $E_{ca}$ = −1.1 V to screen other parameters for this pulsed electrosynthesis. Initially, 0.1 mmol of phenylboronic acid (**1a**, as the model substrate) and 2.0 mmol of

NaNO$_2$ are added into the cathodic cell that contains 0.25 M PBS electrolyte and MeOH. As a test, a 2 s pulse at $E_{an}$ = 0.4 V is followed by a 2 s pulse at $E_{ca}$ = −1.1 V (Supplementary Fig. 7 and Supplementary Note 5), and the sequence is repeated for 5 h. After the reaction is finished, we detect the production of 0.2 mmol of NH$_3$ from the reaction system (Supplementary Fig. 8). However, no phenylamine (**2a**) product is formed. Then, potentiostatic electrolysis was performed for 1 h before beginning the pulsed electrolysis as described above, and **2a** was produced in an 11% isolated yield. It has been reported that rich NH$_3$ is significant for achieving high yields and selectivity of arylamines in the reaction of arylboronic acids with NH$_3$[32,33]. That is, the accumulation of NH$_3$ is necessary for the subsequent coupling of **1a** with NH$_3$. Thus, we conducted continuous electrolysis of 2.0 mmol of NO$_2^-$ at −1.1 V for 7 h before pulsed electrolysis. Time-dependent transformations indicate that nearly full conversion of NO$_2^-$ is finished, and 1.80 mmol of NH$_3$ is finally obtained (Supplementary Fig. 9 and Supplementary Note 6). The massive formation of NH$_3$ provides a guarantee for subsequent C − N coupling to arylamines.

Further screening results reveal that the conversion of **2a** shows a volcanic trend with altering $E_{an}$ and $t_{an}$. A maximum 72% yield of **2a** is obtained at $E_{an}$ = 0.4 V, while $t_{an}$ = 2 s is the best interval for the anodic potentials (Fig. 4d and e). We attribute the inferior conversion of **2a** at

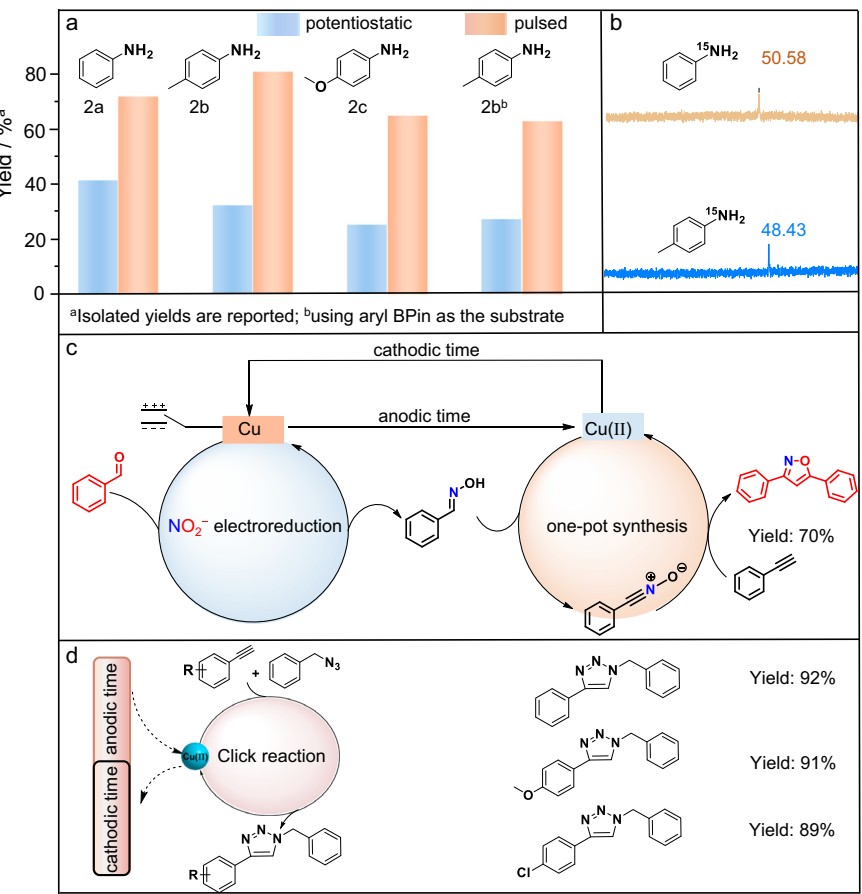

**Fig. 6 | Methodology universality. a** Yields of functionalized primary arylamines by utilizing our pulsed strategy and potentiostatic electrolysis method with additional Cu(II). **b** Two cases of ${}^{15}$N-labelled arylamines with their ${}^{15}$N NMR spectra. **c** One-pot synthesis of 3,5-diphenyl-isoxazole from benzaldehyde with $NO_2^-$ and phenylacetylene by a pulsed electrochemical method. **d** The pulsed electrochemical method for the Click reaction.

lower potentials ($E_{an} < 0.4$ V) and with a shorter $t_{an}$ (1.5 s) to the insufficient formation of Cu(II) under such conditions. Meanwhile, the concentration of Cu(II) ions reflects positive correlations to the applied anodic potentials (Fig. 4f), which may provide support for a higher **2a** yield at a more positive potential. In addition, a further increase in $E_{an}$ and $t_{an}$ is not helpful for the formation of **2a**, which may be due to the increased consumption of NH$_3$ and the electrooxidation of **2a** at $E_{an} = 0.6$ V and with $t_{an} > 2.5$ s. Indeed, the LSV curves of LC-Cu NCs demonstrate that the oxidation of NH$_3$ and **2a** occurs more readily than the OER, and a much lower potential is required for NH$_3$ oxidation than that of **2a** (Supplementary Fig. 10 and Supplementary Note 7). As shown in Supplementary Fig. 11 and Supplementary Note 8, the yield of NH$_3$ decreases with increasing anodic potential. Moreover, we observed a rapid color change of the solution and detected the azobenzene product by gas chromatography–mass spectrometry (GC–MS) when subjecting **2a** to pulsed electrolysis with an $E_{an}$ of 0.6 V (Supplementary Fig. 12 and Supplementary Note 9), indicating the oxidation of **2a**. Therefore, we identify the optimal pulsed conditions: $E_{ca} = -1.1$ V, $E_{an} = 0.4$ V, and $t_{an} = t_{ca} = 2$ s.

**Unveiling the pulsed roles and proposing a possible mechanism**

The nitrite electroreduction reaction (NO$_2^-$RR) to NH$_3$ is an important step in the construction of the C−N bond. Isotope-labelled ${}^1$H NMR spectra were obtained to prove the nitrogen source. The ${}^1$H NMR spectra of the electrolyte after the electrocatalytic reduction of Na${}^{15}$NO$_2$ show the typical double peaks of ${}^{15}$NH$_4^+$ at $\delta = 6.84$ and 7.02 ppm (Supplementary Fig. 13 and Supplementary Note 10), demonstrating that the synthesis of NH$_3$ resulted from the electroreduction of NO$_2^-$. Then, electrochemical in situ ATR-FTIR spectra were obtained to

capture the key intermediates during the NO$_2^-$ reduction process. The peak located at 1254 cm$^{-1}$ is assigned to *NO$_2$ and declines progressively owing to the conversion of NO$_2^-$ as electrolysis proceeds. The peaks located at 1654 cm$^{-1}$ and 1163 cm$^{-1}$ belong to *NO and *NH$_2$, the key intermediates during NO$_2^-$ reduction. In addition, these peaks shift to lower wavenumbers for *${}^{15}$NO (1594 cm$^{-1}$) and *${}^{15}$NH$_2$ (1093 cm$^{-1}$) due to the isotope effect (Supplementary Fig. 14 and Supplementary Note 11).

Control experiments were carried out to further investigate the roles of the pulsed potential technique. Cu(II) is well known as the key catalytic species for Cham-Lam coupling reactions[20,21,42]. The ultraviolet–visible (UV–vis) fluorescence spectra show the presence of Cu(II) ions during the reaction (Fig. 5a), which is due to the electrooxidation-induced leaching of Cu(II) at the pulsed $E_{an}$. In contrast, the comparison experiment in which the electricity was turned off after potentiostatic conditions for 7 h to reduce NO$_2^-$ to NH$_3$ was tested, and only 9% aniline was generated, far less than the concentration of aniline after pulsed electrolysis (Fig. 5d). This control experiment further indicates the importance of the pulsed step to form Cu(II) in situ. The UV–vis spectra show that there exists more Cu(II) with a higher $E_{an}$. Thus, higher oxidation potentials are required for oxidizing Cu and leaching hypervalent Cu(II) species to catalyze C − N couplings. This may account for why the yield of **2a** increases as $E_{an}$ increases until $E_{an} = 0.4$ V. Then, the coordination of Cu(II) with NH$_3$ is essential for triggering subsequent transformations[21]. The formed Cu(II)-NH$_3$ complex is further confirmed by electron paramagnetic resonance (EPR) tests (Fig. 5b), and a blue color is also displayed owing to the dissolution of the complex in the solution (Supplementary Fig. 15 and Supplementary Note 12). Furthermore, in situ Raman spectra reveal that Cu$_2$O and CuO can be formed on the Cu electrode surface with

increasing oxidation potential (Fig. 5c). This may account for why the yield of **2a** increases as $E_{an}$ increases until $E_{an} = 0.4$ V. Most importantly, Cu(II) can be readily removed from the solution by electroplating when the reaction is complete to ensure that the Cu level meets the emission standards before safe disposal (Supplementary Fig. 15 and Supplementary Note 12). The expedient recycling of Cu(II) not only avoids the waste of Cu metals but also reduces the detriment of Cu residuals to the products and environment. The pH value of the solution becomes 11.8 when the reaction is finished, which is lower than that of the one-pot $NO_2^-$ to **2a** by adding extra Cu(II) and under potentiostatic conditions (pH = 13.5, Fig. 5d and Supplementary Table 2). The much slower increase in pH in our system is mainly attributed to the electrooxidation of Cu ($2Cu + 2OH^- - 2e^- \rightarrow Cu_2O + H_2O$, $Cu_2O + 2OH^- - 2e^- \rightarrow 2CuO + H_2O$, $Cu + 2OH^- - e^- \rightarrow CuO + H_2O$)[29,30] and ammonia at $E_{an}$ (Ref. 36), causing the consumption of $OH^-$. As a result, the phenol byproduct is suppressed (Fig. 5d, Supplementary Fig. 16, and Supplementary Note 13). Compared to potentiostatic electrolysis at $-1.1$ V, Cu(II) can remain stable for a longer period under pulsed conditions (Supplementary Fig. 17 and Supplementary Note 14), offering a great opportunity for achieving long-term electrolysis mediated by molecular metal catalysts. Arylboronic acid is easily activated by $OH^-$ ions to form nucleophilic $PhB(OH)_3^-$ (Ref. 49), which will migrate toward the anode during anodic time owing to electrostatic attraction. This will mitigate mass transport limitations and facilitate the interaction of $PhB(OH)_3^-$ with the Cu(II)·$NH_3$ complex, thereby speeding up C–N bond formation. Importantly, due to the rapid polarity switch, the concentrations of Cu(II) and $NH_3$ are high near the electrode surface, thus promoting Chan-Lam coupling. Finally, the overall reaction scheme is described in Fig. 5e. Electroreduction of $NO_2^-$ first occurs over the surface of the LC-Cu cathode to generate $NH_3$. When the potential is switched to $E_{an}$, Cu(II) is generated and then quickly coordinates with $NH_3$ in the EDL to form the Cu(II)·$NH_3$ intermediate. This intermediate further reacts with the activated $PhB(OH)_3^-$ that migrates to the EDL to yield **2a** and release the Cu(II) catalyst. Cu(II) either participates in the next catalytic cycle or is deposited on the electrode surface when applying a cathodic potential. Furthermore, the XRD patterns, XPS spectra, and Cu LMM Auger electron spectroscopy (AES) spectra of LC-Cu NCs reveal that copper oxides are formed on the surface of Cu after pulsed electrolysis (Supplementary Fig. 18 and Supplementary Note 15). The formed copper oxides can also be reduced to form Cu(0) to repeat the whole reaction process.

## Applications to other reactions

On the basis of the above understandings, we evaluate the generality of our method in the synthesis of functionalized primary arylamines from $NO_2^-$ with different arylboronic acids. A series of arylboronic acids with both electron-withdrawing and electron-donating substituents on the aryl ring are all amenable to our strategy, producing the corresponding amines in good yields. Aryl Bpins can also replace arylboronic acids to produce corresponding anilines with good yields under standard conditions, suggesting substrate applicability (Supplementary Table 3, Supplementary Figs. 20–31 and Supplementary Notes 16–27). Delightedly, the yields of these products are all higher than those using the one-pot procedure by requiring an additional Cu(II) source and under constant potential conditions (Fig. 6a), showing the high efficiency of our method. Additionally, $^{15}N$-labelled arylamines, which have demonstrated potential applications in the preparation of $^{15}N$-labelled drugs for studying their metabolic profiles[50], can be facilely synthesized by using low-cost and easy-to-handle $Na^{15}NO_2$ as the $^{15}N$ source (Fig. 6b), demonstrating more economic and safety advantages than using $^{15}NH_3$. A $^{15}N$ abundance of 98.99% in the product of the $^{15}N$-labelled compound was obtained (Supplementary Fig. 19). In addition, the method can be applied to a copper ion-catalyzed cycloaddition reaction for the one-pot two-step

synthesis of 3,5-diphenyl-isoxazole by utilizing in situ generated benzaldoxime and phenylacetylene, namely, (i) the electroreduction of $NO_2^-$ to $NH_2OH$, followed by benzaldehyde to benzaldoxime over an LC-Cu NC cathode, and (ii) subsequent synthesis of 3,5-diphenyl-isoxazole from benzaldoxime with phenylacetylene catalyzed by in situ generated Cu(II) (Fig. 6c, Supplementary Fig. 32 and Supplementary Note 28). Furthermore, the pulsed electrochemical protocol can be applied to other reactions of terminal alkynes with benzyl azides to construct 1,2,3-triazole N-heterocycles (Fig. 6d, Supplementary Figs. 33–35 and Supplementary Notes 29–31), implying good methodology universality.

## Discussion

In summary, a pulsed electrochemistry method is developed to enable the one-pot heterogeneous/homogeneous cascade catalytic transformation of $NO_2^-$ and arylboronic acids to primary arylamines by employing LC-Cu as the cathode in an aqueous solution. The whole reaction process involves first reductive hydrogenation of $NO_2^-$ to $NH_3$ over a $Cu_2O$-derived low-coordinated Cu nanocoral cathode and tandem C–N coupling of $NH_3$ with arylboronic acids catalyzed by in situ dissolved Cu(II) via electrooxidation at a switched anodic potential to deliver arylamines in the same reactor. The Cu(II) species and important Cu(II)·$NH_3$ intermediate are validated by in situ Raman, UV–vis, and EPR experiments. In addition, compared with potentiostatic electrolysis by adding additional Cu(II), our pulsed process can also mitigate pH alternation, thus suppressing phenol byproducts and increasing arylamine selectivity. The rapid alternating polarity leads to increasing concentrations of $NH_3$, $ArB(OH)_3^-$, and Cu(II) near the Cu electrode surface, further improving the reaction efficiency. Furthermore, the pulsed protocol enables a reversible shift between the release of Cu(II) on the anodic pulse and the deposition of Cu(II) on the cathodic pulse, promoting long-term electrosynthesis. Our strategy is well applied to the synthesis of different functionalized arylamines. Impressively, $^{15}N$-labelled arylamines are expediently synthesized by using $Na^{15}NO_2$, avoiding the use of an expensive $^{15}NH_3$ source. Importantly, the pulsed electrochemical protocol-induced leaching of Cu(II) ions can be further utilized to catalyze the cycloaddition reaction to synthesize 3,5-diphenyl-isoxazole and click reactions with a reverse deposition of Cu(II) at the cathodic potential. Our strategy does not need an additional Cu(II) source and increases the durability via easy deposition recovery, contributing to green synthesis. In the future, we will focus on developing a pulsed electrochemical method for continuous synthesis by using a flow cell and further optimizing the catalyst and reaction parameters to increase the reaction rate. Our work represents an advance in electrocatalytic C–N bond construction from inorganic nitrogen sources, which not only provides a promising alternative for the synthesis of primary arylamines with $NO_2^-$ as the nitrogen source[51–53] and in situ generated Cu(II) as catalysts but also offers a paradigm for other electrochemical transformations for boosting the reaction efficiency/selectivity and achieving a more durable electrosynthesis.

## Methods
### Materials
All chemicals were analytical grade and used as received without further purification. Deionized water was used in all experiments.

### Synthesis of self-supported Cu(OH)₂ nanowires (NWAs)
$Cu(OH)_2$ NWAs were synthesized according to the reported literature[54]. A piece of commercial Cu foam (3 cm × 1 cm × 0.1 cm) was ultrasonically treated with acetone and 3.0 M HCl solution for 15 min. Then, the Cu foam was washed with distilled water (DW). Under magnetic stirring, NaOH (3.0 g, 0.075 mmol) and $(NH_4)_2S_2O_8$ (0.68 g, 0.003 mmol) were dissolved in 30 mL DI water, and the precleaned Cu substrate was immersed into the mixed solution. The Cu substrate

covered with blue $Cu(OH)_2$ was removed from the solution after 15 min, followed by washing with deionized water and drying at room temperature.

### Synthesis of self-supported $Cu_2O$ nanocorals (NCs)

The $Cu(OH)_2$ NWAs were placed into a porcelain boat in a tube furnace and heated for 2 h at 500 °C at a heating rate of 1 °C/min in an Ar atmosphere. The final product was cooled to room temperature under an Ar atmosphere to obtain $Cu_2O$ NCs.

### In situ electroreduction of $Cu_2O$ NCs to LC-Cu NCs

The Cu NCs were prepared in situ in a typical divided three-electrode system using 0.25 M PBS (pH = 5.8) and MeOH (2:1 $v/v$) as the electrolyte. The as-prepared $Cu_2O$ NCs with an exposed surface area of 1.0 cm$^2$ served as the working electrode, a Pt plate was used as the counter electrode, and Hg/HgO was used as the reference electrode. Then, LC-Cu NCs were fabricated through facile electroreduction of $Cu_2O$ at −0.8 V vs. Hg/HgO until the reductive peaks fully disappeared.

### Characterization

Scanning electron microscopy (SEM) images were taken with an FEI Apreo S LoVac scanning electron microscope. Transmission electron microscopy (TEM) images were obtained with a JEOL-2100F system equipped with EDAX Genesis XM2. The in situ X-ray diffraction (XRD) measurements were performed on a Rigaku Smartlab9KW Diffraction System using monochromated Cu Kα radiation. X-ray photoelectron spectrum (XPS) measurements were determined by a photoelectron spectrometer using Al Kα radiation as the excitation source (PHI 5000 VersaProbe). All the peaks were calibrated with the C 1s spectrum at a binding energy of 284.8 eV. In situ Raman spectrum measurements were performed on a Renishaw inVia reflex under excitation with 532 nm laser light. The NMR spectra were recorded on Varian Mercury Plus 400 instruments at 400 MHz ($^1$H NMR) and 101 MHz ($^{13}$C NMR). Chemical shifts were reported in parts per million (ppm) downfield from internal tetramethylsilane. Multiplicity was indicated as follows: s (singlet), d (doublet), t (triplet). Coupling constants were reported in hertz (Hz). Gas chromatography–mass spectrometry (GC–MS) was carried out with a TRACE DSQ. The ultraviolet–visible (UV–Vis) absorbance spectra were measured on a Beijing Purkinje General T6 new century spectrophotometer.

### General procedure for pulsed potential enabled one-pot cascade transformation of $NO_2^-$ and arylboronic acids to arylamines in an aqueous electrolyte

All purchased chemicals were used directly without further treatment. Linear sweep voltammetry (LSV) and chronoamperometry were performed using an electrochemical workstation (CS150H). The electrochemical experiments were carried out in a divided three-electrode H-type cell, which was separated into a cathode cell and an anode cell by the NF117 membrane. The as-prepared catalyst was taken as the working electrode (exposure area of 1.0 cm$^2$), Pt was used as the counter electrode, and Hg/HgO was used as the reference electrode. Unless otherwise specified, a mixed solution of 0.25 M PBS and MeOH (2:1 $v/v$, 24 mL) was used as an electrolyte. After Cu NCs were formed in situ, 0.1 mmol of phenylboronic acid and 2.0 mmol of $NaNO_2$ were added into the cathode cell and sonicated to form a homogeneous solution. Then, electrochemical experiments were carried out under potentiostatic conditions at −1.1 V vs Hg/HgO for 7 h to reduce nitrite to ammonia and performed under pulsed potential conditions at different $E_{ca}$ and $E_{an}$ values and with different $t_{ca}$ and $t_{an}$ values for 5 h. During the pulsed electrolysis process, the OER occurs at the cathodic time, and the HER occurs at the anodic time for the counter electrode. The reaction process was monitored by thin-layer chromatography (TLC) or GC–MS. After completion of the reaction, the product was extracted with dichloromethane (DCM), and the combined organic

extracts were washed with brine, dried over anhydrous magnesium sulfate, filtered, and concentrated in vacuo. The crude product was purified by column chromatography on silica gel or using a TLC plate, and the corresponding aniline was obtained as a colorless oil. All experiments were carried out at room temperature (RT, 25 ± 0.5 °C).

### General procedure for pulsed potential-enabled Click reaction

The experiments were carried out in a divided three-electrode H-type cell consisting of an LC-Cu NC working electrode, a Pt plate counter electrode, and a Hg/HgO reference electrode, which was separated into a cathode cell and an anode cell by the membrane. The mixed solution of 0.25 M PBS and MeOH (2:1 $v/v$, 7.5 mL) was used as the electrolyte. After the LC-Cu NCs were formed in situ, 0.12 mmol of benzyl azide, 0.1 mmol of aryl alkynes, and 0.04 mmol of sodium ascorbate were added into the cathode and sonicated to form a homogeneous solution. Then, electrochemical experiments were performed under pulsed potential conditions ($E_{ca}$ = −0.4 V, $E_{an}$ = 0.6 V, $t_{ca}$ = 1 s, $t_{an}$ = 2 s) until aryl alkynes were completely consumed. Then, the product was extracted with DCM, and the combined organic extracts were washed with brine, dried over anhydrous magnesium sulfate, filtered, and concentrated in vacuo. The crude product was purified by column chromatography on silica gel or using a TLC plate. All experiments were carried out at room temperature (RT, 25 ± 0.5 °C).

### General procedure for the one-pot synthesis of 3,5-disubstituted isoxazoles

A typical experimental procedure for the synthesis of 3,5-disubstituted isoxazoles was carried out in a divided three-electrode H-type cell. Meanwhile, the LC-Cu, Pt plate, and a Hg/HgO electrode were used as the working electrode (the working area is 1.0 cm$^2$), the counter electrode, and the reference electrode, respectively. The mixed solution of 0.5 M PBS and t-BuOH (2:1 $v/v$, 24 mL) was utilized as the electrolyte. Then, 0.2 mmol of benzaldehyde and 2.0 mmol of $NaNO_2$ were added into the cathode cell and sonicated to form a homogeneous solution. The electrochemical experiments were performed under potentiostatic conditions at −0.7 V vs Hg/HgO until benzaldehyde was completely consumed. After that, chloramine-T·3$H_2O$ (0.2 mmol) was added in several portions, followed by phenylacetylene (0.2 mmol) and 0.04 mmol of sodium ascorbate, and the electrochemical experiments were performed under pulsed potential conditions ($E_{ca}$ = −0.7 V, $E_{an}$ = 0.6 V, $t_{ca}$ = 1 s, $t_{an}$ = 2 s) until phenylacetylene was completely consumed.

### In situ Raman spectroscopy measurements

The in situ Raman measurements were performed on the aforementioned Raman microscope and electrochemical workstation in the previous paragraph. The electrochemical cell was homemade by Teflon with a thin round quartz glass plate as a cover. The working electrode was immersed in the electrolyte through the battery wall, and the electrode surface was kept perpendicular to the laser. Raman spectra were acquired under different electrochemical conditions.

### Determination of ammonia-N concentration

An ultraviolet–visible (UV–Vis) spectrophotometer was used to detect the ion concentration of posttest electrolytes after diluting them to the appropriate concentration to match the range of calibration curves. Ammonia-N was determined using Nessler's reagent as the color reagent. A certain amount of electrolyte was removed from the electrolytic cell and diluted to 5 mL to the detection range. Next, 0.1 mL potassium sodium tartrate solution ($\rho$ = 500 g L$^{-1}$) was added and mixed thoroughly, and then 0.1 mL Nessler's reagent was added to the solution. The absorption intensity at a wavelength of 420 nm was recorded after sitting for 20 min. The concentration-absorbance curve was calibrated using a series of standard ammonium chloride

solutions, and the ammonium chloride crystal was dried at 105 °C for 2 h in advance.

## $^{15}$N-labelling experiment

Na$^{15}$NO$_2$ (99.21%) was used as the $^{15}$N source for the labelling experiments. $^{15}$N-labelling experiments were carried out in a divided three-electrode H-type cell composed of a Cu NC working electrode (1.0 cm$^2$), a Pt plate counter electrode, and a Hg/HgO reference electrode. Then, 0.1 mmol of phenylboronic acid and 2.0 mmol of Na$^{15}$NO$_2$ were added into the cathode cell for an electrochemical reaction. After the reaction was finished, the electrolyte solution was extracted by DCM and purified using a TLC plate. The obtained products were characterized by GC–MS to determine the source of nitrogen.

## UV–Vis spectroscopy measurements of Cu(II)

The UV–Vis spectrum was recorded at 452 nm (maximum wavelength absorbance of Cu(II) complex). Then, 2.0 mL of the reaction solution was added to 5 mL of DDTC-Na solution (54 μg mL$^{-1}$) and diluted to 10 mL. Tests were carried out after 10 min of stability. A blank analysis was performed before any measurements. The full range wavelength scan of DDTC-Na in 0.25 PBS and MeOH mixed solution was taken before the addition of any reagents: no peak at 452 nm. After adding the reaction solution, a peak appeared and grew in intensity at 452 nm.

## EPR experiments for trapping the Cu(II)·NH$_3$ complex

X-band EPR spectra of Cu(II)·NH$_3$ were recorded after the electrochemical reaction (experimental conditions: frequency, 9.4278 GHz; power, 2.0 mW; modulation, 0.8 mT)[21]. After that, the solution was packed into a 2.8 mm outer diameter quartz tube with a height of 2 cm. The data were collected at ambient temperature.

## Product quantification

After the chronoamperometry test finished, the electrolyte solution was extracted with DCM several times. The organic phase containing the aniline products was subjected to GC–MS analysis. The yield (%) of the products was calculated using the equation below. The extracted structures of the products were further confirmed by NMR and mass spectra. Note that each data point in the main context was repeated three times. The error bars are included in the figures.

$$\text{Yield.(\%)(for products) :} = \frac{n(\text{obtained products})}{n(\text{theoretically formed products})} \times 100\% \tag{1}$$

$$\text{FE(\%)} = \frac{Q(\text{theoretically values})}{Q(\text{actual values})} \times 100\% \tag{2}$$

$$\text{TOF(s}^{-1}) = \frac{\text{The amount of produced ammonia molecules per s}}{\text{The amount of Cu sites}} \tag{3}$$

## Data availability

The data that support the plots within this paper are available from the corresponding author upon reasonable request. The source data underlying Figs. 3–6 are provided as a Source Data file. Source data are provided with this paper.

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

## Acknowledgements

We acknowledge the National Natural Science Foundation of China (grant no. 22271213 to B.Z., grant no. 22101202 to Y.W., and grant no. 22001192 to C.L.) and the Postdoctoral Research Foundation of China (grant no. 2022M722357 to Y.W.) for financial support.

## Author contributions

B.Z. conceived the idea and directed the research. M.H. and Y.W. carried out the experiments and analysed the experimental results. M.L., R.L. and Y.W. contributed to the discussion. C.L. and B.Z. wrote the paper with the comments of all the authors.

## Competing interests

The authors declare no competing interests.
