## [Peer Review File · Nature Communications]

Aqueous pulsed electrochemistry promotes C–N bond formation via a one-pot cascade approachREVIEWER COMMENTS

Reviewer #1 (Remarks to the Author):

In this manuscript He and co-workers describe electrochemical amination of arylboronic acid derivatives using nitrite as amine source. The method features an interesting application of pulsed electrolysis into electrochemical synthesis as well as utilisation of nitrite which is not a common amine source for Chan-Lam coupling. However, the overall goal of this work is ambiguous as utilisation of NO₂/NO₃ is claimed to be for environmental reason (meaning that such reaction needs to be done on enormous scale to be impactful) in the introduction, yet the main utility of Chan-Lam coupling is in fine-chemical industry (i.e., production scale is likely to be insignificant to impact environment). Although the authors might have different opinions on this point, it is reasonable to say that chances to conduct Chan-Lam coupling to process nitrogen waste for alleviate environmental impact would be slim. I think it is fair to say that the justification for combining NO₂ reduction and Chan-Lam coupling is a force-fit. I recommend authors to choose either solving nitrogen waste problem or developing better Chan-Lam reaction. Solutions for these two problems are usually mutually exclusive and cannot be solved at the same time due to massively different operation scale. With the current form it is difficult to find any use in either of the scenario.

Reviewer #2 (Remarks to the Author):

In the submitted article, the authors report a pulsed electrocatalysis method to reduce NO₂⁻ to NH₃, and subsequently couple NH₃ with arylboronic acids to produce arylamines. Under pulsed reaction conditions NH₃ was produced from NO₂⁻ on a Cu cathode under reductive potentials, while the Cu(II) was formed in-situ under oxidative potentials and served as the species to catalyze the coupling between NH₃ and arylboronic acids. The authors used UV-vis and EPR to prove the formation of Cu(II) and Cu(II)-NH₃, which are the key species involved in the construction of arylamines. Further, this pulsed electrolysis method can also inhibit the formation of phenol by-product by accelerating the migration of nucleophilic ArB(OH)₃⁻, which is the intermediate that reacts with Cu(II)-NH₃ directly. The work presents an facile method to produce arylamines, and the concept of “combining Cu heterogeneous catalysts and Cu(II) homogenous catalyst” is one that is not often explored. Despite these unique aspects, the main limitation of the work is that it is essentially system for the reduction NO₂⁻ to NH₃, without a deep dive into the mechanism of this reaction. For example, there is no quantification of Faradaic efficiency for NH₃, although the yield is noted.

Practically speaking, I am not convinced the method is practical. What is the advantage of doing the pulsed step if NH₃ can just be produced electrochemically and combined with a Cu(II) catalyst in a second step? It seems to be that the pulsing to reduce and oxidize Cu is just an unnecessary expenditure of energy, and plenty of highly active NO_x reduction catalysts already exist. Because of this, I would not recommend publishing in an impactful journal at the level of Nature Communications.

Minor comments are given below.

The authors should include a discussion on the abundance and availability/cost of NO₂⁻

Page 3 Line 45 and Line 49, spelling mistake: “surces” should be “sources” and “cross-couling” should be “cross-coupling”.

Page 8 Figure 3b, authors labeled peaks at 220, 625, and a peak around 420 cm^{-1} , but in Line 147, they mentioned the peaks located 148, 220, and 625 cm^{-1} . Why authors do not label 148 in the figure and not mention the peak around 420 cm^{-1} on Line 147?

Figure 3c, is it possible for the authors to use some symbol to label the peak of Cu and Cu₂O on each XRD pattern directly? It's difficult to see the Cu and Cu₂O peaks of the JCPDS card clearly.

In Figure 3d, the authors conducted XPS to show the valence state of Cu₂O and Cu NCs, although there is a slight difference between these two samples, actually, it's difficult to use XPS to distinguish Cu(I) or Cu(0), Auger has a better sensitivity to confirm the valence state of Cu(I) or Cu(0).

In Figure 3f and lines 164-165, the description of EXAFS data is unclear. The author said the coordination shell of LC-Cu is lower than Cu foam. What are the exact coordination numbers of LC-Cu and Cu foam? Page 10, Line 200, the author conducted 7h electrolysis of NO₂⁻ before pulsed electrolysis, and they think the nearly full conversion of NO₂⁻ is finished. How to confirm it's finished? It's better to plot the yield of NH₃ with changes in the time, to make sure the production of NH₃ continues to increase during the past 7h and decreases after 7h.

For Figure 4d and 4e, what's the exact condition of the electrolysis? From 5h pulse electrolysis there's no phenylamine (2a), then authors conducted potentiostatic electrolysis before pulse electrolysis, so in Figure 2d and 2e, did the author conduct potentiostatic electrolysis for 7h to obtain enough NH₃, and then conducted pulse electrolysis? There's no detailed description of the Methods part, SI, or legend under the figures. And for the yield of phenylamine, what's the yield of the other by-product? And what are the other by-products? Only phenol, or include azobenzene? Page 11, with a higher anode potential, the yield of 2a decrease, and authors attributed it to the oxidation of NH₃ and 2a. It's better to provide the production of NH₃ (without arylboronic acids) at different anodic potential, to prove that the yield of NH₃ decrease at higher anodic potential. And for the oxidation of 2a, the author prove that at 0.6V there exist azobenzene, is it possible for author provide the production of azobenzene under different E_{an} (also different tan) to confirm the result?

Figure 5b, EPR plot, what do the different curves stand for? Cu(II)+ NH₃ is the solution after pulse electrolysis or the mixture of commercial Cu(II) and commercial NH₃? Describe the detail. And the blue color of the solution might be the Cu²⁺ ion, not the Cu(II) complex, the color is not adequate evidence for proving there exist Cu(II)-NH₃ complex.

In Figure 5c, the authors conducted in-situ Raman, and try to use in-situ Raman to prove the significance of higher oxidation potentials. But CuO and Cu(II) are not the same things, the authors already use Figure 4f to prove that with a higher E_{an}, there exists more Cu(II), no need to use in-situ Raman. And

from in-situ Raman, under 0.2V, there is no CuO, but from Figure 4f, there exists Cu(II) in the solution below 0.2V, and the results are not consistent.

Figure S8, lacks the detail of the experiments. Did the author add Cu(II) when they start the electrolysis experiment? Cu(II) will be reduced during the reaction, and when adding Cu(II), no need to conduct the pulse electrolysis, because there already exists Cu(II) for Chan-Lam reaction, no need to add potential. And the pH for pulse electrolysis is 11.8, when authors compare the pH effect of the formation of phenol, why did they choose pH 5.8 instead of 11.8?

Mechanism: What is the reaction pathway or equation to form Ph-OH by-product? And there is no description of the intermediate in NO₂⁻ reduction reaction. It's a scheme to describe the reaction process (similar to Figure 2d), not a mechanism.

Is fig. 4b normalized to geometric surface area? To show intrinsic differences in activity, I would recommend normalizing to real surface area (e.g. through UPD or capacitance measurements) The authors used 0.25 M PBS and MeOH, with the addition of NaNO₂ and phenylboronic acid as the electrolyte (page 9, line 189). The use of PBS is reasonable since acidic media promote HER and basic media result in undesired byproducts like phenol, so neutral pH is selected. On the other hand, the role of MeOH needs more elaboration.

The authors mention that MeOH can inhibit HER to boost the NO₂⁻ electroreduction, and MeOH is ideal solvent for coupling boric acid with NH₃. There is no mention of the role of MeOH in the mechanism at all. As an example, please look at the (Scientific Reports, 2021, 11, 12185). Figure 5d clearly shows that aniline is produced even without the electricity input. Therefore, it is recommended to report the product distribution data with and without MeOH.

The authors point out to the Cham-Lan reaction, but also Mitsunobu reaction might occur, since there is a phosphate, a phenol, an alcohol, and a metal catalyst present in the system. Distinction from other reaction systems is recommended in the introduction section, for general readers.

The authors successfully explained the bright sides of the proposed system, but obviously it is not fully the case. Since the authors used phrases like "Massive formation of NH₃" (page 10, line 201) or "...showing the high efficiency of our method." (page 14, line 290), or "...demonstrating more economic and safety advantages than using 15NH₃." (page 14, line 294);

To show the possible directions for the future research, it is recommended to discuss a little about the problems facing the system. As an example, please look at (Energy Fuels, 2022, 36, 12737–12749).

Figure 4b compares the LSV between Cu foam and the LC-Cu. The 140 mV written on the graph can be misleading. In general, recommended to report the turnover frequency for comparing materials with different morphologies and crystal structures. SEM images in Supplementary Figure 1 clearly show the

need of TOF for comparison.

Also, the X-axis ascending direction in Figure 4c is different with the other figures.

It seems there is a typo in page 11, line 242. The intended potential might be -0.2 V, not the positive 0.2.

Reviewer #3 (Remarks to the Author):

He and colleagues describe the coupling of C–N bonds catalyzed by copper using a pulsed electrochemistry technique. This methodology represents a green approach to the synthesis of arylamines from in-situ nitrite reduction with subsequent coupling to arylboronic acids, without the need for an added copper catalyst. A mechanism is proposed with support for the intermediates, and the authors demonstrate how the use of the pulsed electrolysis method allows for a more stable copper catalyst. The generality of the method was shown by the use of substituted boronic acids, as well as ^{15}N labeled nitrite, and the application of the process to Click chemistry is very exciting. Reducing nitrite to ammonia and using it in-situ without the need for added copper represents a novel and sustainable approach to the synthesis of aryl amines, with potential for industrial scale implementation. Therefore, this manuscript is suitable for publication in Nature Communications. Prior to publication, the following minor comments should be addressed.

1. In the abstract, the phrase “wide substrate scope” is used, however the authors only used arylamines with three different substituents, all in the para position. In order to make this claim, the authors should add more examples, especially with groups of an electron-withdrawing nature or in various positions around the aryl ring.
2. The sentence from lines 37-43 on page 3 is a bit long, consider breaking it into two for a more readable introduction.
3. In the introduction, the authors describe several approaches for forming C – N bonds, and use the phrase “impeding methodology development and applications” (line 50, page 3), though they do not make it obvious why that is the case. Adding a sentence to explain what is wrong with current methods, or why the use of an NH_3 surrogate is undesirable, would help with setting up the challenges solved by the present method.
4. A few comments on Figure 2. Figure 2a seems unnecessary, I’m not sure what it adds that isn’t already explained in Figure 1/the text. A better replacement for this Figure would be a depiction of the overall cell reaction(s), since there is no mention in the text of what is happening at the counter electrode. Similarly, I’m not sure why Figure 2c is necessary given that the reactions taking place during the cathodic time are depicted in Figure 2e.
5. For Figure 3a, a simple label on the top and bottom images (Cu_2O and Cu , for example) would help the reader.
6. Page 9, the sentence in lines 171–173 is confusing.
7. In multiple instances, such as page 9 line 188, the authors mention the reaction yield, but never the Faradaic Efficiency (FE). This metric should be reported, and could easily be added to some of the plots

such as Figures c–e.

8. Page 9 line 188, E_{an} should be E_{cat}.

9. How necessary is the pulsed electrolysis following the electrochemical reduction of nitrite? Although it seems obvious that the leached Cu would be needed for the coupling reaction, a control experiment wherein the reaction is run under potentiostatic conditions for 7 hours to reduce NO₂⁻ to NH₃, and then the electricity is turned off for the duration of the reaction, would be beneficial.

10. The phrase “more easily oxidized than the OER” (page 11 line 223 and page S4) is awkward, consider rewording to “... demonstrate that the oxidation of NH₃ and 2a occurs more readily than OER”.

11. Page 11 line 235, the authors should give references to support the statement that Cu(II) is a well known intermediate in Chan Lam coupling.

12. Were these reactions run under N₂, or air? If so, the control experiment proving that the N does come from NO₂⁻ reduction and not the less likely N₂ reduction would be beneficial, especially for the synthesis of ¹⁵N labeled compounds since the %¹⁵N is not indicated.

13. Page 15 line 317 should be “shift” not “swift”.

14. For the general procedure beginning on page 17 line 366, the authors do not mention that the electrolysis was run under potentiostatic conditions first for 7 hours to reduce nitrite to ammonia, however this is mentioned in text. If this 7 hour electrolysis was commonly conducted, it should be stated in the general procedure.

15. On a similar note, the authors should mention what material was used as a counter electrode and membrane for the general procedure, as well as how long the pulsed electrolysis was run for if the time or charge was kept constant.

A point-by-point response to the reviewers' comments

To reviewer 1:

Reviewer letter: In this manuscript He and co-workers describe electrochemical amination of arylboronic acid derivatives using nitrite as amine source. The method features an interesting application of pulsed electrolysis into electrochemical synthesis as well as utilisation of nitrite which is not a common amine source for Chan-Lam coupling. However, the overall goal of this work is ambiguous as utilisation of NO_2/NO_3 is claimed to be for environmental reason (meaning that such reaction needs to be done on enormous scale to be impactful) in the introduction, yet the main utility of Chan-Lam coupling is in fine-chemical industry (i.e., production scale is likely to be insignificant to impact environment). Although the authors might have different opinions on this point, it is reasonable to say that chances to conduct Chan-Lam coupling to process nitrogen waste for alleviate environmental impact would be slim. I think it is fair to say that the justification for combining NO_2 reduction and Chan-Lam coupling is a force-fit. I recommend authors to choose either solving nitrogen waste problem or developing better Chan-Lam reaction. Solutions for these two problems are usually mutually exclusive and cannot be solved at the same time due to massively different operation scale. With the current form it is difficult to find any use in either of the scenario.

Answer: We appreciate the reviewer for giving a positive comment on our work *“The method features an interesting application of pulsed electrolysis into electrochemical synthesis as well as utilization of nitrite which is not a common amine source for Chan-Lam coupling.”* The formation of electrochemically coupled C-N bonds driven by renewable energy sources is emerging as a promising sustainable strategy.

In this manuscript, we only selected the classical Chan-Lam coupling as a model reaction to demonstrate electrocatalytic C-N bond construction from inorganic nitrogen sources. Our work not only provides a promising alternative for the synthesis of primary arylamines with NO_2^- as the nitrogen source and in situ generated Cu(II) as catalysts but also offers a paradigm for other electrochemical transformations for boosting the reaction efficiency/selectivity and achieving more durable electrosynthesis. Furthermore, the pulsed electrochemical protocol can be expanded to the Click reactions of terminal alkynes with benzyl azides to construct 1,2,3-triazole N-heterocycles, implying good methodology universality.

We apologize that the description in the introduction may lead to a misunderstanding, and we have modified our introduction as follows: *“The construction of C-N bonds has attracted much attention due to nitrogen-containing compounds being widely used in chemical and pharmaceutical industries as well as in materials science. However, high-temperature and high-pressure reaction conditions lead to serious*

environmental pollution problems; thus, it is important to explore new synthetic strategies for C–N bond formation. Recently, the formation of electrochemically coupled C–N bonds driven by renewable energy sources has emerged as a promising sustainable strategy...”.

In this manuscript, we report a pulsed electrochemistry-promoted one-pot hetero/homogeneous cascade catalytic transformation of NO_2^- and arylboronic acids to primary arylamines in an aqueous solution. The significant points of our manuscript are listed below:

(1) Pulsed potentials enabled cascade conversion of NO_2^- and arylboronic acids to arylamines via NO_2^- electroreduction to NH_3 over a Cu cathode followed by in situ formed Cu(II)-catalyzed C–N coupling of NH_3 with arylboronic acids.

- ✓ NH_3 and Cu(II) are both generated in situ via NO_2^- electroreduction and oxidative dissolution of the Cu electrode, **avoiding direct use of NH_3 and the addition of a Cu catalyst.**
- ✓ Our pulsed electrolysis shows approximately **twice the yield of traditional potentiostatic electrolysis**, which requires additional Cu(II) for the synthesis of arylamines.

(2) Combining in situ and quasi-in situ spectroscopy to unveil the roles of the pulsed potential technique.

- ✓ In situ Raman and UV–vis spectra reveal the formation of Cu(II) under a pulsed anodic 0.4 V, and electron paramagnetic resonance confirms the key Cu(II)- NH_3 complex for C–N coupling.
- ✓ Pulsed electrolysis can slow the pH value increase compared with potentiostatic electrolysis, thus intrinsically inhibiting the phenol byproduct.
- ✓ Cu(II) can remain stable for a longer period under pulsed conditions due to periodic dissolution of the Cu electrode and redeposition of Cu(II), offering a great chance for long-term electrosynthesis.

(3) Outstanding methodology universality and promising utility.

- ✓ Different functionalized arylboronic acids and aryl BPin are amenable to our strategy, delivering the corresponding primary arylamines in good yields.
- ✓ ^{15}N -labelled arylamines as key building blocks for ^{15}N -labelled drugs are economically synthesized by using $\text{Na}^{15}\text{NO}_2$, avoiding the use of an expensive $^{15}\text{NH}_3$ source.
- ✓ The pulsed protocol can be expanded to the Click reaction for fabricating 1,2,3-triazole molecules, showing good applicability.

Our work of pulse-promoted electrosynthesis of amines represents an advance in electrocatalytic C–N bond construction from inorganic nitrogen sources, which not only provides a promising alternative for the synthesis of primary arylamines by use of NO_2^- instead of direct contact with NH_3 and in situ generated Cu(II) as catalysts but also offers a new paradigm for other electrochemical transformations for boosting the reaction efficiency/selectivity and achieving a more durable electrosynthesis.

We highly appreciate the reviewer’s thorough reading and constructive comments/suggestions about our manuscript!

To reviewer 2:

Reviewer letter: In the submitted article, the authors report a pulsed electrocatalysis method to reduce NO_2^- to NH_3 , and subsequently couple NH_3 with arylboronic acids to produce arylamines. Under pulsed reaction conditions, NH_3 was produced from NO_2^- on a Cu cathode under reductive potentials, while Cu(II) was formed in situ under oxidative potentials and served as the species to catalyze the coupling between NH_3 and arylboronic acids. The authors used UV-vis and EPR to prove the formation of Cu(II) and Cu(II)- NH_3 , which are the key species involved in the construction of arylamines. Further, this pulsed electrolysis method can also inhibit the formation of phenol by-product by accelerating the migration of nucleophilic $\text{ArB}(\text{OH})_3^-$, which is the intermediate that reacts with Cu(II)- NH_3 directly. The work presents an facile method to produce arylamines, and the concept of “combining Cu heterogeneous catalysts and Cu(II) homogenous catalyst” is one that is not often explored. Despite these unique aspects, the main limitation of the work is that it is essentially system for the reduction NO_2^- to NH_3 , without a deep dive into the mechanism of this reaction. For example, there is no quantification of Faradaic efficiency for NH_3 , although the yield is noted. Practically speaking, I am not convinced the method is practical. What is the advantage of doing the pulsed step if NH_3 can just be produced electrochemically and combined with a Cu(II) catalyst in a second step? It seems to be that the pulsing to reduce and oxidize Cu is just an unnecessary expenditure of energy, and plenty of highly active NO_x reduction catalysts already exist. Because of this, I would not recommend publishing in an impactful journal at the level of Nature Communications.

Answer: We highly appreciate the reviewer for recognizing the uniqueness of our work “*combining Cu heterogeneous catalysts and Cu(II) homogenous catalyst is one that is not often explored*”. However, for the reviewer’s other comments, we need to explain them as follows. To save the reviewer’s valuable time, key revisions are displayed in a yellow background in the revised manuscript and Supplementary Materials. We are sure that the quality of this work will be greatly improved after being revised.

First, we report a pulsed electrochemistry-promoted one-pot hetero/homogeneous cascade catalytic transformation of NO_2^- and arylboronic acids to primary arylamines in an aqueous solution. Our work focuses on advancing electrocatalytic C–N bond construction from inorganic nitrogen sources, which not only provides a promising alternative for the synthesis of primary arylamines with NO_2^- as the nitrogen source and in situ generated Cu(II) as catalysts but also offers a paradigm for other electrochemical transformations for boosting the reaction efficiency/selectivity and achieving more durable electrosynthesis. In addition, the reduction of NO_2^- to NH_3 indeed has a great influence on the subsequent C–N bond construction, and our group and others have reported many works about the reduction of $\text{NO}_2^-/\text{NO}_3^-$ to NH_3 (*Natl. Sci. Rev.* 2019, 6, 730-738; *J. Am. Chem. Soc.* 2020, 142, 7036-7046; *Chem. Soc. Rev.* 2021, 50, 6720-6733), and we have carried out supplementary experiments including the Faradaic efficiency (FE) for NH_3 and the in situ ATR-FTIR spectra and others on the mechanism of the

NO_2^- reduction reaction and provided a point-by-point response.

Then, the advantages of the pulsed step are listed below:

(1) Pulsed potentials enabled cascade conversion of NO_2^- and arylboronic acids to arylamines via NO_2^- electroreduction to NH_3 over a Cu cathode followed by in situ formed Cu(II)-catalyzed C–N coupling of NH_3 with arylboronic acids. $\text{Cu}(\text{OAc})_2$ has become the “classic” Cu source for Chan–Lam reactions. Additives/ligands are often needed. In our work, pulsed electrochemistry can produce Cu(II) via in situ electrooxidation of the Cu electrode rather than extra addition and does not require additional ligands to stabilize Cu(II), which is favorable to lower the cost and simplify the operations.

(2) The pulsed method avoids the use of molecular oxygen or organic oxidants. Currently, most reported Chan-Lam systems rely on molecular oxygen or organic oxidants. In addition, the electrochemical setup is proposed to assist oxidative turnover (*J. Am. Chem. Soc.* 2021, 143, 6257–6265). In our system, a pulsed electrochemical system that can form Cu(II) in situ and maintain high concentrations of Cu(II) without molecular oxygen or other chemical oxidants offers a great opportunity for achieving long-term electrolysis mediated by homogeneous metal catalysts. To further verify the benefits of the pulsed step, $\text{Cu}(\text{OAc})_2$ was added into the system after NO_2^- electroreduction to NH_3 . As shown in Figure R1, the yield of aniline is only 55%, less than that obtained by pulse electrolysis, which proves that the pulsed method can facilitate the construction of the C–N bond.

Fig. R1 Comparison experiment by adding $\text{Cu}(\text{OAc})_2$. Conditions: $\text{Ph-B}(\text{OH})_2$ (0.1 mmol), NaNO_2 (2 mmol), and 0.03 mmol $\text{Cu}(\text{OAc})_2$ were added and reacted for 5 h without electricity after NO_2^- electroreduction to NH_3 at -1.1 V .

(3) Pulsed electrochemistry inhibits the formation of the byproduct phenol by adjusting the pH of the solution. Previous reports proved that phenylboronic acid was easy to hydrolyse into phenol in an aqueous solution (*Tetrahedron Lett.* 2003, 44, 1691-1694), and the phenol content increased with increasing pH, which was further confirmed by our experiment. Therefore, most of the previous reports required strict control of anhydrous conditions, which hindered practical application. In our work, pulsed electrochemistry causes the consumption of OH^- near the cathode surface because of the electrooxidation of Cu ($2\text{Cu} + 2\text{OH}^- - 2\text{e}^- \rightarrow \text{Cu}_2\text{O} + \text{H}_2\text{O}$, $\text{Cu}_2\text{O} + 2\text{OH}^- - 2\text{e}^- \rightarrow 2\text{CuO} + \text{H}_2\text{O}$, $\text{Cu} + 2\text{OH}^- - \text{e}^- \rightarrow \text{CuO} + \text{H}_2\text{O}$) at E_{an} , which can slow the change in pH, thus accelerating C–N formation and suppressing phenol byproducts.

Thus, we believe that pulsed electrolysis is a powerful tool for promoting electrocatalytic C–N coupling reactions. In fact, we only selected the classical Chan-Lam coupling as a model reaction to demonstrate electrocatalytic C–N bond construction from inorganic nitrogen sources. Our work not only provides a promising alternative for the synthesis of primary arylamines with NO_2^- as the nitrogen source and in situ generated Cu(II) as catalysts but also offers a paradigm for other electrochemical transformations for boosting the reaction efficiency/selectivity and achieving more durable electrosynthesis. The methodological expansion to the Click reaction highlights its great promise.

Several days ago, I was notified that a pulsed method for similar C–N construction from the other group was requested to be revised by *Nat. Chem.* As one of the reviewers, I provided a very positive evaluation on their *Nat. Chem.* manuscript work (MS ID: NCHEM-221225xx) although we have some competing conflicts. As we know, electrocatalytic C–N bond formation from inorganic nitrogen wastes is an emerging sustainable method to fabricate valuable organic C–N chemicals. I think we should have a big picture in promoting the healthy development of the C–N electrochemical field.

Comment 1: There is no quantification of Faradaic efficiency for NH_3 , although the yield is noted.

Answer: Thank you for pointing out this issue. The quantification of FE for NH_3 under different reduced potentials has been provided in the revised manuscript (Fig. 4c). As the result shows, -1.1 V is optimal for NO_2^- electroreduction, giving rise to NH_3 with the highest yield and FE.

Fig. R2 Potential-dependent yields and FEs of NH_3 via NO_2^- reduction over LC-Cu in the mixed 0.25 M PBS and MeOH (2:1 v/v) solution.

Comment 2: The authors should include a discussion on the abundance and availability/cost of NO_2^-

Answer: Thank you for the reviewer's comments. Nitrite (NO_2^-) is a vital intermediate oxyanion of the nitrogen cycle that is highly soluble in water and is one of the main contaminants in aquatic ecosystems, and human health. Moreover, NO_2^- can be produced by the oxidation of N_2 through plasma,

photo/electro catalysis (*J. Am. Chem. Soc.* 2022, 144, 10193-10200). NO_2^- is a desirable N-source for NH_3 electrosynthesis from the standpoint of NH_3 synthesis and environmental conservation, especially because NO_2^- has a low bond dissociation energy (*Chem. Rev.* 2009, 109, 2209-2244). We have added a discussion on the abundance in the revised manuscript "And as a sustainable alternative to traditional thermal and enzymatic catalysis, the synthesis of $\text{NO}_3^-/\text{NO}_2^-$ by N_2 oxidation followed by C-N bond construction using $\text{NO}_3^-/\text{NO}_2^-$ electroreduction techniques will push it to a new level due to the frequent invocation of C-N construction in synthetic chemistry".

Comment 3: Page 3 Line 45 and Line 49, spelling mistake: "surces" should be "sources" and "cross-couling" should be "cross-coupling".

Answer: We acknowledge the reviewer's kind suggestion. We have changed "surces" and "cross-couling" to "sources" and "cross-coupling" in the revised manuscript. We have also carefully checked the whole manuscript for typos and grammar errors, and the revisions are displayed on a yellow background in the revised manuscript.

Comment 4: Page 8 Figure 3b, authors labeled peaks at 220, 625, and a peak around 420cm^{-1} , but in Line 147, they mentioned the peaks located 148, 220, and 625cm^{-1} . Why authors do not label 148 in the figure and not mention the peak around 420cm^{-1} on Line 147?

Answer: Thank you for pointing out this issue. The peaks located at 148, 220, 420 and 625cm^{-1} belong to Cu_2O . We have labelled them in the figure and mentioned them in the revised manuscript (Fig. 3b).

Comment 5: Figure 3c, is it possible for the authors to use some symbol to label the peak of Cu and Cu_2O on each XRD pattern directly? It's difficult to see the Cu and Cu_2O peaks of the JCPDS card clearly.

Answer: Thank you for the reviewer's kind suggestions. We have used different symbols to label the peaks of Cu and Cu_2O on the XRD pattern in the revised manuscript.

Comment 6: In Figure 3d, the authors conducted XPS to show the valence state of Cu_2O and Cu NCs, although there is a slight difference between these two samples, actually, it's difficult to use XPS to distinguish Cu(I) or Cu(0), Auger has a better sensitivity to confirm the valence state of Cu(I) or Cu(0).

Answer: Thank you for the suggestion. We have added Auger spectra of Cu_2O and Cu NCs to better identify Cu(I) or Cu(0) in the revision (Fig. 3d). In addition, the presence of Cu(II) in the Cu NC Auger

spectrum may be due to the oxidation of the material during testing.

Fig. R3 Cu LMM AES spectra of Cu_2O and Cu NCs.

Comment 7: In Figure 3f and lines 164-165, the description of EXAFS data is unclear. The author said the coordination shell of LC-Cu is lower than Cu foam. What are the exact coordination numbers of LC-Cu and Cu foam?

Answer: To better investigate the local coordination environment of LC-Cu and Cu foam, we fitted the EXAFS data. The results show that the average Cu–Cu coordination number of LC-Cu (9.75) is significantly lower than that of Cu foam, which has also been added to the revised manuscript and Supplementary Materials (Supplementary Fig. 4 and Table 1).

Table R1. EXAFS fitting parameters at the Cu K-edge for various samples ($S_0^2=0.837$)

Sample	Path	C.N.	R (Å)	σ^2 (Å ²)	ΔE (eV)	R factor
Cu foam	Cu-Cu	12*	2.54±0.01	0.0085	4.34±0.6	0.0052
LC-Cu	Cu-Cu	9.75±0.8	2.54±0.01	0.0094	3.64±0.9	0.0131

^aC.N. : coordination numbers; ^bR: bond distance; ^c σ^2 : Debye-Waller factors; ^d ΔE : the inner potential correction.

Fig. R4 **a** Fourier transform (FT) to R-space to isolate the EXAFS contributions from each coordination shell. **b** The $\chi(k)$ data weighted by k^3 . The quantified fitting results are shown in Table R1.

Comment 8: Page 10, Line 200, the author conducted 7h electrolysis of NO_2^- before pulsed electrolysis, and they think the nearly full conversion of NO_2^- is finished. How to confirm it's finished? It's better to plot the yield of NH_3 with changes in the time, to make sure the production of NH_3 continues to increase during the past 7h and decreases after 7h.

Answer: After considering the kind suggestion, we have added a plot of the yield of NO_2^- and NH_3 with changes over time in the revised manuscript (Supplementary Fig. 9). Time-dependent transformations indicate that the yield of NO_2^- continuously decreases with prolonged reaction time, while the yield of NH_3 continues to increase until 7 h and basically remains unchanged, indicating that the conversion of NO_2^- is nearly complete.

Fig. R5 Time-dependent amount of substance change of NO_2^- and NH_3 over LC-Cu NCs at -1.1 V.

Comment 9: For Figure 4d and 4e, what's the exact condition of the electrolysis? From 5 h pulse electrolysis there's no phenylamine (2a), then authors conducted potentiostatic electrolysis before pulse electrolysis, so in Figure 2d and 2e, did the author conduct potentiostatic electrolysis for 7 h to obtain enough NH_3 , and then conducted pulse electrolysis? There's no detailed description of the Methods part, SI, or legend under the figures.

Answer: Thank you for the reviewer's comments. For figures 2d and 2e, we conducted potentiostatic electrolysis for 7 h to obtain enough NH_3 and then conducted pulse electrolysis. Relevant experimental details have been added to the general procedure of the revised manuscript.

Comment 10: And for the yield of phenylamine, what's the yield of the other by-product? And what are the other by-products? Only phenol, or include azobenzene? Page 11, with a higher anode potential, the yield of 2a decrease, and authors attributed it to the oxidation of NH_3 and 2a. It's better to provide the production of NH_3 (without arylboronic acids) at different anodic potential, to prove that the yield of NH_3 decrease at higher anodic potential. And for the oxidation of 2a, the author prove that at 0.6V there

exist azobenzene, is it possible for author provide the production of azobenzene under different E_{an} (also different t_{an}) to confirm the result?

Answer: Thank you for the reviewer's suggestions. First, the byproducts include phenol and azobenzene. We have added the yields of phenol and azobenzene under different E_{an} and t_{an} in the revised manuscript. The results show that azobenzene exists at 0.6 V due to the oxidation of **2a**. In addition, we found increased phenol content at higher E_{an} and longer t_{an} , which may be due to the oxidation of phenylboronic acid (*J. Org. Chem.* 2013, 78, 7482-7487). Then, we added the production of NH_3 (without arylboronic acids) at different anodic potentials, which can prove that the yield of NH_3 decreases at higher anodic potentials (Supplementary Fig. 11).

Fig. R6 **a** Isolated yields of **2a** and byproducts under pulsed electrolysis conditions with $E_{ca} = -1.1$ V, different E_{an} values, and $t_{an} = t_{ca} = 2$ s. **b** Isolated yields of **2a** and byproducts under pulsed electrolysis conditions: $E_{ca} = -1.1$ V, $t_{ca} = 2$ s; $E_{an} = 0.4$ V, the different t_{an} values. **c** Yield of NH_3 under pulsed electrolysis conditions with $E_{ca} = -1.1$ V, different E_{an} values, and $t_{an} = t_{ca} = 2$ s.

Comment 10: Figure 5b, EPR plot, what do the different curves stand for? $Cu(II) + NH_3$ is the solution after pulse electrolysis or the mixture of commercial $Cu(II)$ and commercial NH_3 ? Describe the detail. And the blue color of the solution might be the Cu^{2+} ion, not the $Cu(II)$ complex, the color is not adequate evidence for proving there exist $Cu(II)-NH_3$ complex.

Answer: Thank you for pointing out this issue. For the EPR plot, $Cu(II) + NH_3$ is the solution after pulse electrolysis. The relevant experimental details have been added under the legend of the figure in the revised manuscript. Additionally, thanks for the valuable suggestion, the color is not adequate evidence for proving that there exists a $Cu(II)-NH_3$ complex. We combined the color and the EPR plot to demonstrate that the solution contains a $Cu(II)-NH_3$ complex in the revised supporting information.

Comment 11: In Figure 5c, the authors conducted in-situ Raman, and try to use in-situ Raman to prove the significance of higher oxidation potentials. But CuO and $Cu(II)$ are not the same things, the authors

already use Figure 4f to prove that with a higher E_{an} , there exists more Cu(II), no need to use in-situ Raman. And from in-situ Raman, under 0.2V, there is no CuO, but from Figure 4f, there exists Cu(II) in the solution below 0.2V, and the results are not consistent.

Answer: Thank you for the reviewer's wise comments. We apologize for our misleading description. As the reviewer said, CuO and Cu(II) are indeed two types of Cu species. The Cu electrode can be oxidized to Cu(II) leached into the electrolyte and CuO under high oxidation potentials (*Angew. Chem. Int. Ed.* 2021, 60, 7418-742). Combining the results of UV-vis and in situ Raman, we find that Cu(II) forms more easily than CuO. The UV-vis spectra also show that Cu(II) can be generated at $E_{an} = -0.4$, and more Cu(II) exists with a higher E_{an} . When the potential further increases to 0.2 V, the CuO species formed on the Cu electrode surface, which was revealed by in situ Raman spectroscopy. The relevant descriptions have been corrected in the revised manuscript.

Comment 12: Figure S8, lacks the detail of the experiments. Did the author add Cu(II) when they start the electrolysis experiment? Cu(II) will be reduced during the reaction, and when adding Cu(II), no need to conduct the pulse electrolysis, because there already exists Cu(II) for Chan-Lam reaction, no need to add potential. And the pH for pulse electrolysis is 11.8, when authors compare the pH effect of the formation of phenol, why did they choose pH 5.8 instead of 11.8?

Answer: Thank you for the comments. The experiments were carried out by adding 0.03 mmol $\text{Cu}(\text{OAc})_2 \cdot \text{H}_2\text{O}$ under different pH conditions without electricity. We have added the details of Supplementary Fig. 8 and the contrast experiment under pH 11.8 in the revised supporting information.

Fig. R7 Yields of phenol byproduct under different pH conditions without electricity (0.01 mmol of phenylboronic acid (**1a**), 0.03 mmol of $\text{Cu}(\text{OAc})_2 \cdot \text{H}_2\text{O}$ as the catalyst, 0.25 M PBS (pH = 5.8, 11.8 and 13.2) and MeOH (2:1 v/v).

Comment 13: Mechanism: What is the reaction pathway or equation to form Ph-OH by-product?

Answer: Lam demonstrated that the presence of water in the system causes oxidation, a competitive Chan–Lam C–O bond formation that uses water as the heteroatomic nucleophile (*Tetrahedron Lett.* 2003, 44, 1691-1694). We have added the pathway in the revised supporting information (Supplementary Fig. 1).

Comment 14: And there is no description of the intermediate in NO_2^- reduction reaction. It's a scheme to describe the reaction process (similar to Figure 2d), not a mechanism.

Answer: Thank you for the comment. We have added the isotope-labelling electrochemical in situ ATR-FTIR spectra of LC-Cu NCs using $^{14}\text{NO}_2^-$ and $^{15}\text{NO}_2^-$ that can capture the intermediates and describe the process of NO_2^- reduction in the revised manuscript and supporting information (Supplementary Fig. 14). As the results show, the peak located at 1254 cm^{-1} is assigned to $^*\text{NO}_2$ and declines progressively owing to the conversion of NO_2^- as electrolysis proceeds. The peaks located at 1654 cm^{-1} and 1163 cm^{-1} belong to $^*\text{NO}$ and $^*\text{NH}_2$, which are the key intermediates during NO_2^- reduction. Additionally, these shifted to lower wavenumbers for ^{15}NO (1594 cm^{-1}) and $^{15}\text{NH}_2$ (1093 cm^{-1}) due to the isotope effect.

Fig. R8 Isotope-labelling electrochemical in situ ATR-FTIR spectra of LC-Cu NCs using $^{14}\text{NO}_2^-$ and $^{15}\text{NO}_2^-$.

Comment 15: Is fig. 4b normalized to geometric surface area? To show intrinsic differences in activity, I would recommend normalizing to real surface area (e.g. through UPD or capacitance measurements).

Answer: Thank you for the kind suggestion. We have added the ECSA of Cu foam and LC-Cu NCs, and the lsv of Figure 4b has been normalized to the geometric surface area in the revised manuscript and supporting information. After normalization by the electrochemical surface area (ECSA), LC-Cu NCs still

possess the highest performance for NO_2^- reduction, implying their high intrinsic activity. The ECSA of Cu foam and LC-Cu NCs has been added to Supplementary Fig. 5.

Fig. R9 CV curves of **a** LC-Cu NC and **b** Cu foam with various scan rates from 20 to 60 mV s^{-1} . **c** The double layer capacitance (C_{dl}) of different catalysts. **d** LSV curves of LC-Cu NCs and Cu foam at a scan rate of 5 mV s^{-1} in a mixed solution of 0.25 M PBS and MeOH (2:1 v/v) with 2.0 mmol of NO_2^-

$$A_{\text{ECSA}}^{\text{Cu NCs}} = \frac{5.4 \text{ mF cm}^{-2}}{40 \mu\text{F cm}^{-2} \text{ per } 2 \text{ cm}^2_{\text{ECSA}}} = 135 \text{ cm}^2_{\text{ECSA}}$$

$$A_{\text{ECSA}}^{\text{Cu foam}} = \frac{4.01 \text{ mF cm}^{-2}}{40 \mu\text{F cm}^{-2} \text{ per } 2 \text{ cm}^2_{\text{ECSA}}} = 100.25 \text{ cm}^2_{\text{ECSA}}$$

Comment 16: The authors used 0.25 M PBS and MeOH, with the addition of NaNO_2 and phenylboronic acid as the electrolyte (page 9, line 189). The use of PBS is reasonable since acidic media promote HER and basic media result in undesired byproducts like phenol, so neutral pH is selected. On the other hand, the role of MeOH needs more elaboration. The authors mention that MeOH can inhibit HER to boost the NO_2^- electroreduction, and MeOH is ideal solvent for coupling boric acid with NH_3 . There is no mention of the role of MeOH in the mechanism at all. As an example, please look at the (Scientific Reports, 2021, 11, 12185). Figure 5d clearly shows that aniline is produced even without the electricity input. Therefore, it is recommended to report the product distribution data with and without MeOH.

Answer: Previous reports have demonstrated that the water dissociation process (the Volmer step) is strongly inhibited due to the hydrogen bonding interaction between methanol and water molecules;

thus, the HER is largely suppressed in the presence of methanol (*ACS Energy Lett.* 2021, 6, 3844–3850). The rate of migration of water molecules to the catalyst surface slows down as the hydrogen bonds between methanol and water molecules become stronger (*J. Am. Chem. Soc.* 2021, 143, 29, 10940–10947). In addition, we have added related experiments to verify the role of methanol in our system. The results show that the HER is obviously suppressed and thus boosts the conversion of NO_2^- to NH_3 , which will further improve the yield of aniline.

Fig. R10 **a** LSV curves of LC-Cu NCs at a scan rate of 5 mV s^{-1} in a solution of 0.25 M PBS with MeOH or without MeOH. **b** Yields and FEs of NH_3 via NO_2^- reduction over LC-Cu in a solution of 0.25 M PBS with MeOH or without MeOH. **c** Yields of aniline and phenol over LC-Cu in a solution of 0.25 M PBS with MeOH or without MeOH.

Comment 17: The authors point out to the Cham-Lan reaction, but also Mitsunobu reaction might occur, since there is a phosphate, a phenol, an alcohol, and a metal catalyst present in the system. Distinction from other reaction systems is recommended in the introduction section, for general readers.

Answer: We acknowledge the reviewer for pointing out this issue. The Mitsunobu reaction is an alcohol substitution reaction that uses an acidic pronucleophile, as well as PPh_3 and an azodicarboxylate as reagents. A key stage of the reaction mechanism involves forming alkoxyphosphonium, an in situ activation step by a lone pair on P of PPh_3 that enables subsequent $\text{S}_{\text{N}}2$ displacement with the nucleophile along with inversion of stereochemistry (*Nat. Chem.* 2019, 11, 966–967). In our system, the PBS buffer solution consists of K_2HPO_4 and KH_2PO_4 , which do not have a long pair on P and cannot catalyze the Mitsunobu reaction.

Comment 18: The authors successfully explained the bright sides of the proposed system, but obviously it is not fully the case. Since the authors used phrases like “Massive formation of NH_3 ” (page 10, line 201) or “...showing the high efficiency of our method.” (page 14, line 290), or “...demonstrating more economic and safety advantages than using $^{15}\text{NH}_3$.” (page 14, line 294); To show the possible directions for the future research, it is recommended to discuss a little about the problems facing the system. As an example, please look at (*Energy Fuels*, 2022, 36, 12737–12749).

Answer: Thank you for the reviewers' comment. We have revised our description to make it easier for readers to accept in the revised manuscript. We have added a discussion about the problem in our system as follows: "In the future, we should develop the pulsed electrochemical method for continuous synthesis by using the flow cell and further optimize the reaction parameters to increase the reaction rate".

Comment 19: Figure 4b compares the LSV between Cu foam and the LC-Cu. The 140 mV written on the graph can be misleading. In general, recommended to report the turnover frequency for comparing materials with different morphologies and crystal structures. SEM images in Supplementary Figure 1 clearly show the need of TOF for comparison.

Answer: Thank you for the kind suggestion. We have added the TOF of the conversion of NO_2^- to NH_3 and the NH_3 yield rate over LC-Cu NCs and Cu foam in the revised version. The results show that the performance of the electroreduction of NO_2^- to NH_3 over LC-Cu NCs is much better than that over Cu foam.

Fig. R11 **a** CV curves of LC-Cu NCs and Cu foam at a scan rate of 5 mV s^{-1} . **b** TOF values and **c** yield rate and FE of NH_3 via NO_2^- reduction over LC-Cu NCs and Cu foam in a mixed solution of 0.25 M PBS and MeOH (2:1 v/v) at -1.1 V .

Comment 20: Additionally, Also, the X-axis ascending direction in Figure 4c is different with the other figures.

Answer: Thank you for the reviewer's comments. We have changed the X-axis ascending direction of Figure 4c in the revised manuscript.

Comment 21: It seems there is a typo in page 11, line 242. The intended potential might be -0.2 V , not the positive 0.2 .

Answer: Thank you for the comment from the reviewer. In situ Raman spectra reveal that CuO appears when the potential of E_{an} reaches 0.2 V after careful checking.

Department of Chemistry
Tianjin University
Tianjin 300072, P. R. China
Tel&Fax: 86-22-27403475
E-mail: bzhang@tju.edu.cn

We highly appreciate the reviewer's thorough reading and constructive comments/suggestions about our manuscript!

To reviewer 3:

Reviewer letter: He and colleagues describe the coupling of C–N bonds catalyzed by copper using a pulsed electrochemistry technique. This methodology represents a green approach to the synthesis of arylamines from in-situ nitrite reduction with subsequent coupling to arylboronic acids, without the need for an added copper catalyst. A mechanism is proposed with support for the intermediates, and the authors demonstrate how the use of the pulsed electrolysis method allows for a more stable copper catalyst. The generality of the method was shown by the use of substituted boronic acids, as well as ^{15}N labeled nitrite, and the application of the process to Click chemistry is very exciting. Reducing nitrite to ammonia and using it in-situ without the need for added copper represents a novel and sustainable approach to the synthesis of aryl amines, with potential for industrial scale implementation. Therefore, this manuscript is suitable for publication in Nature Communications. Prior to publication, the following minor comments should be addressed.

Answer: We highly appreciate the reviewer for the positive comments on our manuscript. Regarding the concerns or comments of the reviewer, we have provided point-by-point responses. To save the reviewer's valuable time, key revisions are displayed on a yellow background in the revised manuscript and Supporting information. We are sure that the quality of this work will be greatly improved after being revised.

Comment 1: In the abstract, the phrase "wide substrate scope" is used, however the authors only used arylamines with three different substituents, all in the para position. In order to make this claim, the authors should add more examples, especially with groups of an electron-withdrawing nature or in various positions around the aryl ring.

Answer: Thank you for the reviewer's comments. After considering the kind suggestions of the reviewer, we now apply **another 10 examples of aryl substrates**, including electron-withdrawing groups and groups in various positions around the aryl ring. For instance, the selected aryl substrates all work well under our electrochemistry system, giving rise to the desired arylamines with good yields. These new data have been added to the revised supporting information (Supplementary Table 3).

Table R2 Yields of functionalized primary arylamines by utilizing our pulsed strategy. ^[a]Condition: Ar-B(OH)₂ (0.1 mmol), NaNO₂ (2 mmol), 0.25 M PBS:MeOH (24 mL), LC-Cu NCs (working area: 1 cm²), air, rt. Isolated yields are reported. ^[b] Using Ar-BPin as the substrate

Comment 2: The sentence from lines 37-43 on page 3 is a bit long, consider breaking it into two for a more readable introduction.

Answer: Thank you for pointing out this issue. We have revised the previous expression to a more appropriate form in the revised manuscript for better understanding. The revision is also extracted as follows: “Excitingly, coupling inorganic nitrogen sources (e.g., nitrogen (N₂), NO₃⁻, NO₂⁻) and carbon sources (e.g., carbon dioxide, carbon monoxide, and other carbonyl substrates) to form upgraded organic amines is becoming a rapidly emerging area of electrocatalytic C–N bond construction. As a sustainable alternative to traditional thermal and enzymatic catalysis, the synthesis of NO₃⁻/NO₂⁻ by N₂ oxidation followed by C–N bond construction using NO₃⁻/NO₂⁻ electroreduction techniques will push it to a new level due to the frequent invocation of C–N construction in synthetic chemistry”.

Comment 3: In the introduction, the authors describe several approaches for forming C–N bonds, and use the phrase “impeding methodology development and applications” (line 50, page 3), though they do not make it obvious why that is the case. Adding a sentence to explain what is wrong with current methods, or why the use of an NH₃ surrogate is undesirable, would help with setting up the challenges solved by the present method.

Answer: Thank you for your kind suggestion. We have changed the previous expression to a more appropriate form in the revised manuscript to better understand the significance of our research. Despite these achievements, C–N bond construction mainly starts from small nitrogen and carbon

sources through (i) condensation of hydroxylamine with carbonyl intermediates or highly active carbonyl substrates and (ii) cross-coupling of in situ formed nitrogen and carbon free radical intermediates. However, the current studies are limited in reaction scope, and most of them are small molecule reactions, impeding methodology development and applications. Although electrocatalytic C–N coupling by using additional NH_3 has been reported, the electrocatalytic synthesis of organic amines from NH_3 generated in situ via $\text{NO}_3^-/\text{NO}_2^-$ electroreduction has rarely been studied because the collection and use of NH_3 is relatively complex and has lower reactivity than other nitrogen intermediates.

Comment 4: A few comments on Figure 2. Figure 2a seems unnecessary, I'm not sure what it adds that isn't already explained in Figure 1/the text. A better replacement for this Figure would be a depiction of the overall cell reaction(s), since there is no mention in the text of what is happening at the counter electrode. Similarly, I'm not sure why Figure 2c is necessary given that the reactions taking place during the cathodic time are depicted in Figure 2e.

Answer: Thank you for pointing out this issue. There is a certain degree of similarity between Figure 2a and Figure 1. We have modified Figure 2 as follows in the revised manuscript. The OER occurs at the cathodic time, and the HER occurs at the anodic time for the counter electrode.

Fig. R12 a pulsed potential waveform, **b** NO_2^- reduction before pulsed electrolysis, **c** Cu(II) generation and Chan-Lam coupling under the anodic potential (E_{an}).

Comment 5: For Figure 3a, a simple label on the top and bottom images (Cu_2O and Cu , for example) would help the reader.

Answer: Thank you for the kind suggestion. We have added Cu_2O and LC-Cu to the top and bottom SEM images in the revised manuscript.

Comment 6: Page 9, the sentence in lines 171–173 is confusing.

Answer: Thank you for the kind suggestion. Combined with the comment of reviewer 2, we have explained that the water dissociation process (the Volmer step) is strongly inhibited due to the hydrogen bonding interaction between methanol and water molecules and that the rate of migration of H_2O to the catalyst surface slows down as the hydrogen bonds become stronger; thus, the HER is largely suppressed in the presence of methanol. The relevant description has been added to the revision.

Comment 7: In multiple instances, such as page 9 line 188, the authors mention the reaction yield, but never the Faradaic Efficiency (FE). This metric should be reported, and could easily be added to some of the plots such as Figures c–e.

Answer: Thank you for the kind suggestion. We report a pulsed electrochemistry-promoted one-pot hetero/homogeneous cascade catalytic transformation of NO_2^- and arylboronic acids to primary arylamines. The quantification of FE for NH_3 under different reduced potentials has been provided in the revised manuscript. As the result shows, -1.1 V is optimal for NO_2^- electroreduction, giving rise to NH_3 with the highest yield and 95% FE (Fig. R2). However, Chan-lam coupling involves nonelectrochemical processes that cannot calculate the FE.

Comment 8: Page 9 line 188, E_{an} should be E_{cat} .

Answer: We acknowledge the reviewer for pointing this out. The " E_{an} " has been corrected to " E_{ca} " in the revised manuscript. We have also carefully revised the spelling and grammatical errors in the revised manuscript and supporting information.

Comment 9: How necessary is the pulsed electrolysis following the electrochemical reduction of nitrite? Although it seems obvious that the leached Cu would be needed for the coupling reaction, a control experiment wherein the reaction is run under potentiostatic conditions for 7 hours to reduce NO_2^- to NH_3 and then the electricity is turned off for the duration of the reaction would be beneficial.

Answer: We acknowledge the reviewers' suggestion. We have added a comparison experiment in which the electricity is turned off after potentiostatic conditions for 7 hours to reduce NO_2^- to NH_3 in the

revised manuscript, and the result shows that only 9% aniline is generated, far less than the concentration of aniline after pulsed electrolysis, further indicating the importance of the pulsed step. In addition, the generation of 9% aniline may be due to a small amount of Cu(II) (0.079 mmol) dissolved in an alkaline solution, which was verified by UV-vis spectroscopy.

Fig. R13 Comparisons of the one-pot NO_2^- to **2a** under different conditions, including without electricity, pulsed conditions and potentiostatic conditions with additional Cu(II) after potentiostatic conditions for 7 hours to reduce NO_2^- to NH_3 .

Comment 10: The phrase “more easily oxidized than the OER” (page 11 line 223 and page S4) is awkward, consider rewording to “... demonstrate that the oxidation of NH_3 and **2a** occurs more readily than OER”.

Answer: Thank you for the kind suggestion. The sentence “more easily oxidized than the OER” has been replaced by “ ... demonstrate that the oxidation of NH_3 and **2a** occurs more readily than OER” in the revision.

Comment 11: Page 11 line 235, the authors should give references to support the statement that Cu(II) is a well known intermediate in Chan Lam coupling.

Answer: Thank you for the comment. We have added references that can support that Cu(II) is a well-known intermediate in Chan-Lam coupling in the revised manuscript (*Chem. Rev.* 2019, 119, 12491–12523; *J. Am. Chem. Soc.* 2021, 143, 6257–6265; *J. Am. Chem. Soc.* 2017, 139, 4769–4779).

Comment 12: Were these reactions run under N_2 , or air? If so, the control experiment proving that the N does come from NO_2^- reduction and not the less likely N_2 reduction would be beneficial, especially for the synthesis of ^{15}N labelled compounds since the % ^{15}N is not indicated.

Answer: Thank you for the constructive comment. First, these reactions in our system are both carried out under air. Then, ^{15}N isotope labelling ^1H NMR spectra and in situ ATR-FTIR spectra using $^{14}\text{NO}_2^-$ and

$^{15}\text{NO}_2^-$ have been conducted to prove that the N does come from NO_2^- reduction and not the less likely N_2 reduction in the revised version. The ^1H NMR spectra of the electrolyte after the electrocatalytic reduction of $\text{Na}^{15}\text{NO}_2$ show the typical double peaks of $^{15}\text{NH}_4^+$ at $\delta = 6.84$ and 7.02 ppm, demonstrating that the synthesis of NH_3 resulted from the electroreduction of NO_2^- . The in situ ATR-FTIR also captures the key intermediates during the $^{15}\text{NO}_2^-$ reduction process, which further indicates that the N comes from NO_2^- . Meanwhile, we used HR-MS to calculate that the % ^{15}N of ^{15}N -labelled compounds was 98.99%. These results both indicate that the N source comes from NO_2^- reduction.

Fig. R14 a ^1H NMR spectra of the electrolyte after the electrocatalytic NO_2^- reaction using $^{14}\text{NO}_2^-$ and $^{15}\text{NO}_2^-$ nitrogen sources. b Isotope-labelling electrochemical in situ ATR-FTIR spectra of LC-Cu NCs using $^{15}\text{NO}_2^-$. c ^{15}N abundance of Aniline- ^{15}N .

Comment 13: Page 15 line 317 should be “shift” not “swift”.

Answer: Thank you for pointing out this issue. We have changed “shift” to “swift” in the revised manuscript, and we have carefully examined the full manuscript and supporting information.

Comment 14: For the general procedure beginning on page 17 line 366, the authors do not mention that the electrolysis was run under potentiostatic conditions first for 7 hours to reduce nitrite to ammonia, however this is mentioned in text. If this 7 hour electrolysis was commonly conducted, it should be stated in the general procedure.

Answer: Thank you for the reviewers’ kind suggestion. The electrolysis was carried out under potentiostatic conditions first for 7 hours to reduce nitrite to ammonia and then run under pulsed conditions. Detailed experimental steps have been added to the general procedure in the revised manuscript.

Comment 15: On a similar note, the authors should mention what material was used as a counter electrode and membrane for the general procedure, as well as how long the pulsed electrolysis was run for if the time or charge was kept constant.

Answer: Thank you for the reviewers' kind comments. The electrochemical measurements were carried out in a divided three-electrode electrochemical cell separated by the NF117 membrane. The as-prepared catalyst was taken as the working electrode, Pt was used as the counter electrode, and Hg/HgO was used as the reference electrode. The pulsed electrolysis ended after 5 h. Detailed experimental steps have been added to the general procedure in the revised manuscript.

We highly appreciate the reviewer's thorough reading and constructive comments/suggestions about our manuscript!

REVIEWER COMMENTS

Reviewer #1 (Remarks to the Author):

I appreciate the authors' effort to revise the manuscript. However, importance by connecting NO₂ reduction and Chan-Lam coupling is still not cohesive, and I am not sure what challenges this work is trying to solve. To be fair, each topic - electrochemical NH₃ synthesis, improvement of Chan-Lam coupling and strategic utilization of pulsed electrolysis are all important topics to study. Yet, audiences and challenges for these topics are vastly different. For example, impact of electrochemical NH₃ reduction lies in solving energy challenge created by Harbor-Bosch process and decentralize this energy-intensive process. Downstream utilisation of NH₃ into Chan-Lam reaction is irrelevant in this context, while synthetic chemists have no incentive and reason to swap NH₃, which is convenient and cheap, into electrochemical NO₂ reduction for running Chan-Lam reaction. In fact, Chan-Lam coupling is such an expensive reaction (because boronic acids are expensive) that making simple anilines such as 2a-2b does not justify the value of the transformation (starting material boronic acid is more expensive than these anilines). Same argument for Click reaction: there is no need to generate Cu catalyst by electrochemistry for this reaction. This work rather complicates Click reaction because original simple homogeneous protocol is now heterogeneous system with additional equipment being required. I tried to come up with some suggestions to improve the quality of this work, but it is difficult as each topic is too different. For example, electrochemical NO₂ reduction can be coupled to arene oxidation and produce anilines in a single step via Csp²-H amination fashion, that is certainly something worth considering for Nature Communication as such a reaction could fit into the introduction of this work. However, such suggestion is beyond the scope of this work.

Reviewer #2 (Remarks to the Author):

I agree with the author's justification in promoting fundamental chemistry investigations of C-N coupling, of which there may be translatable aspects over to next-generation future systems. In the end, the authors have taken a lot of steps to improve the manuscript on a technical side and the fundamental interest here would be sufficient in publishing in Nature Communications. I would, however, like to see a nuanced discussion of comparison in the practicality of doing a pulsed electrolysis route, as reported in this paper, vs. a two-step route in nitrate/nitrite electrochemical reduction to NH₃, and combining the electrochemically produced NH₃ with the Cu(II) catalyst and secondary reactants, either in the intro or conclusion. I believe this will be a helpful guide as the electrocatalysis/electrosynthesis community is rapidly growing.

Reviewer #3 (Remarks to the Author):

The authors have addressed most of my previous comments. Some minor comments to consider, related to the previous comments (same number) are listed below.

Comments:

1. It is still a little difficult to tell when Bpin vs. B(OH)₂ was used, why use Bpin in some cases? Is it just to show that you can, in which case, it needs to be more obvious (in manuscript). Is the undecorated aniline example in the manuscript different from the one in the SI? If so, distinguish. Might be more clear to include the examples from the manuscript in Supplementary Table 3 as well.
2. This sentence looks better broken up, however, it would read better if the sentences were shortened altogether, for example: "Coupling inorganic nitrogen sources (...) and carbon sources (...) to form upgraded organic amines is an emerging hot field in electrocatalytic C–N bond construction (refs)."
3. Minor comment, but the part in the sentence "and most of them are small molecule reactions" seems redundant, that can be removed.
4. In the response, the authors mentioned the reactions for the counter electrode (HER and OER) however it still does not seem that this is mentioned in the manuscript or SI. This should either be depicted in the figure, or explained in text in one or both documents. Also, the bottom of Figure 2c is confusing, Cu(II) is consumed during cathodic time, so that part showing the Cu(II) being consumed should be on the bottom of figure 2b. As well, is the OH⁻ related to anodic time, or is the point of showing the OH⁻ just to show that the Ar-OH is not being formed? If it's the latter, I would make this more obvious. I think moving the "Cu(II) consumed" to cathodic time will help make this more obvious.
5. Addressed, thank you.
6. The sentence, as written, still doesn't make much sense, and the point being made here is still a little unclear. I agree with the logic in the sentence that has been added, however I still think the first sentence doesn't seem to have much of a point. As it currently reads, it seems like the first sentence is a random statement of pH-related facts, and the sentence that was added in is explaining why MeOH leads to higher selectivity. It would make more sense to include a short introductory sentence explaining that you're looking at pH and solvent effects, along the lines of "The electrolyte solvent and pH play an important role in reaction selectivity."
7. Addressed, thank you.
8. Addressed, thank you.
9. Addressed, thank you.
10. Addressed, thank you.
11. Addressed, thank you.
12. Addressed, thank you.
13. Addressed, thank you.
14. Mostly addressed, however the authors should indicate the potential that the potentiostatic conditions were run at.
15. Addressed, thank you.

A point-by-point response to the reviewers' comments

To reviewer 1:

Reviewer letter: I appreciate the authors' effort to revise the manuscript. However, importance by connecting NO₂ reduction and Chan-Lam coupling is still not cohesive, and I am not sure what challenges this work is trying to solve. To be fair, each topic - electrochemical NH₃ synthesis, improvement of Chan-Lam coupling and strategic utilization of pulsed electrolysis are all important topics to study. Yet, audiences and challenges for these topics are vastly different. For example, impact of electrochemical NH₃ reduction lies in solving energy challenge created by Harbor-Bosch process and decentralize this energy-intensive process. Downstream utilisation of NH₃ into Chan-Lam reaction is irrelevant in this context, while synthetic chemists have no incentive and reason to swap NH₃, which is convenient and cheap, into electrochemical NO₂ reduction for running Chan-Lam reaction. In fact, Chan-Lam coupling is such an expensive reaction (because boronic acids are expensive) that making simple anilines such as 2a-2b does not justify the value of the transformation (starting material boronic acid is more expensive than these anilines). Same argument for Click reaction: there is no need to generate Cu catalyst by electrochemistry for this reaction. This work rather complicates Click reaction because original simple homogeneous protocol is now heterogeneous system with additional equipment being required. I tried to come up with some suggestions to improve the quality of this work, but it is difficult as each topic is too different. For example, electrochemical NO₂ reduction can be coupled to arene oxidation and produce anilines in a single step via Csp²-H amination fashion, that is certainly something worth considering for Nature Communication as such a reaction could fit into the introduction of this work. However, such suggestion is beyond the scope of this work.

Answer: We appreciate the reviewer for providing a positive comment on improving our work. However, for the importance of our work, we need to explain them as follows:

Our work focuses on advancing electrocatalytic C–N bond construction from inorganic nitrogen sources. Transition metal-mediated C–N bond formation is an essential transformation that enables the preparation of valuable aryl amine products. In recent years, the Chan-Lam coupling reaction has developed rapidly in the field of C–N bond construction, including the coupling of NH₃ with arylboronic acids to form aniline (*J. Am. Chem. Soc.* 2017, 139, 4769–4779). However, the collection, transport, storage, and usage of NH₃ are time- and manpower-consuming processes that require high costs and complex handling. In addition, although the electrochemical reduction of NO₂[−] to NH₃ has been widely reported, the electrosynthesis of organic amines from NH₃ generated in situ via NO₂[−] electroreduction has rarely been studied owing to the lower reactivity over other nitrogen intermediates. In this manuscript, we only selected the classical Chan-Lam coupling as a model reaction to demonstrate

electrocatalytic C–N bond construction from inorganic nitrogen sources (NO_2^-), which not only provides a promising type of reaction for the synthesis of primary aromatics but also offers a paradigm for other electrochemical transformations for boosting the reaction efficiency/selectivity and achieving more durable electrosynthesis.

Our work provides a new method for pulsed electrochemically facilitated catalytic conversion of one-pot heterogeneous/homogeneous cascades. The significant points of the pulsed methods are listed below:

(1) Pulsed potentials enabled cascade conversion of NO_2^- and arylboronic acids to arylamines via NO_2^- electroreduction to NH_3 over a Cu cathode followed by in situ formed Cu(II)-catalyzed C–N coupling of NH_3 with arylboronic acids. $\text{Cu}(\text{OAc})_2$ has become the “classic” Cu source for Chan–Lam reactions. Additives/ligands are often needed. In our work, pulsed electrochemistry can produce Cu(II) via in situ electrooxidation of the Cu electrode rather than extra addition and does not require additional ligands to stabilize Cu(II), which is favorable to lower the cost and simplify the operations. For the Click reaction, although additional Cu(II) homogeneous catalysts can facilitate the reaction, some problems such as long reaction times and difficulties in catalyst recovery can also exist (*Acc. Chem. Res.* 2020, 53, 937–948). Our pulsed electrochemistry can produce Cu(II) via in situ electrooxidation of the Cu electrode, and Cu(II) can be readily removed from the solution by electroplating when the reaction is complete.

(2) The pulsed method avoids the use of molecular oxygen or organic oxidants. Currently, most reported Chan-Lam systems rely on molecular oxygen or organic oxidants. In addition, the electrochemical setup is proposed to assist oxidative turnover (*J. Am. Chem. Soc.* 2021, 143, 6257–6265). However, potentiostatic electrolysis induces irreversible copper plating. In our system, a pulsed electrochemical system that can form Cu(II) in situ and maintain high concentrations of Cu(II) without molecular oxygen or other chemical oxidants offers a great opportunity for achieving long-term electrolysis mediated by homogeneous metal catalysts.

(3) Pulsed electrochemistry improves the selectivity of the products by adjusting the pH of the solution. Previous reports proved that phenylboronic acid was easy to hydrolyse into phenol byproducts in an aqueous solution (*Tetrahedron Lett.* 2003, 44, 1691-1694), and the phenol content increased with increasing pH, which was further confirmed by our experiment. In our work, pulsed electrochemistry causes the consumption of OH^- near the cathode surface because of the electrooxidation of Cu ($2\text{Cu} + 2\text{OH}^- - 2\text{e}^- \rightarrow \text{Cu}_2\text{O} + \text{H}_2\text{O}$, $\text{Cu}_2\text{O} + 2\text{OH}^- - 2\text{e}^- \rightarrow 2\text{CuO} + \text{H}_2\text{O}$, $\text{Cu} + 2\text{OH}^- - \text{e}^- \rightarrow \text{CuO} + \text{H}_2\text{O}$) at E_{an} , which can slow the change in pH, thus accelerating C–N formation and suppressing phenol byproducts.

Thus, we believe that pulsed electrolysis is a powerful tool for promoting electrocatalytic C–N coupling reactions. In fact, we only selected the classical Chan-Lam coupling as a model reaction to demonstrate electrocatalytic C–N bond construction from inorganic nitrogen sources. Our work not only

provides a promising alternative for the synthesis of primary arylamines with NO_2^- as the nitrogen source instead of direct contact with NH_3 and in situ-generated Cu(II) as catalysts but also offers a paradigm for other electrochemical transformations to boost the reaction efficiency/selectivity and achieve more durable electrosynthesis. The methodological expansion to the Click reaction highlights its great promise.

Now, the second and third reviewers gave **positive** comments.

Reviewer 2's evaluation of our work is as follows: "I believe this will be a **helpful guide** as the electrocatalysis/electrosynthesis community is rapidly growing."

The reviewer 3 although "The **generality** of the method was shown by the use of substituted boronic acids.... Reducing nitrite to ammonia and using it in situ without the need for added copper represents a **novel and sustainable** approach to the synthesis of aryl amines, with potential for industrial scale implementation."

Now, we address all the doubts and concerns of the three reviewers. I hope the first reviewer can satisfy our revision.

We highly appreciate the reviewer's thorough reading and constructive comments/suggestions about our manuscript!

To reviewer 2:

Reviewer letter: I agree with the author's justification in promoting fundamental chemistry investigations of C-N coupling, of which there may be translatable aspects over to next-generation future systems. In the end, the authors have taken a lot of steps to improve the manuscript on a technical side and the fundamental interest here would be sufficient in publishing in Nature Communications. I would, however, like to see a nuanced discussion of comparison in the practicality of doing a pulsed electrolysis route, as reported in this paper, vs. a two-step route in nitrate/nitrite electrochemical reduction to NH_3 , and combining the electrochemically produced NH_3 with the Cu(II) catalyst and secondary reactants, either in the intro or conclusion. I believe this will be a helpful guide as the electrocatalysis/electrosynthesis community is rapidly growing.

Answer: We highly appreciate the reviewer's positive comments on our manuscript. We have added a discussion of pulsed electrochemistry in the revised manuscript, as listed below:

"Our pulsed method can produce Cu(II) via in situ electrooxidation of the Cu electrode rather than extra addition, which is favorable for lowering the cost and simplifying the operation procedure. Then, a pulsed electrochemical setup can also assist oxidative turnover and maintain high concentrations of Cu(II) without additional ligands since potentiostatic electrolysis induces irreversible copper plating, which offers a great opportunity for achieving long-term electrolysis mediated by a homogeneous metal catalyst. In addition, pulsed electrochemistry can obviously suppress the formation of phenol byproducts by adjusting the pH of the solution."

To save the reviewer's valuable time, key revisions are displayed on a yellow background in the revised manuscript. We are sure that the quality of this work will be greatly improved after being revised.

We highly appreciate the reviewer's thorough reading and constructive comments/suggestions about our manuscript!

To reviewer 3:

Reviewer letter: The authors have addressed most of my previous comments. Some minor comments to consider, related to the previous comments (same number) are listed below.

Answer: We highly appreciate the reviewer for the positive comments on our manuscript. Regarding the concerns or comments of the reviewer, we have provided point-by-point responses. To save the reviewer's valuable time, key revisions are displayed on a yellow background in the revised manuscript and Supporting information. We are sure that the quality of this work will be greatly improved after being revised.

Comment 1: It is still a little difficult to tell when Bpin vs. B(OH)₂ was used, why use Bpin in some cases? Is it just to show that you can, in which case, it needs to be more obvious (in the manuscript). Is the undecorated aniline example in the manuscript different from the one in the SI? If so, distinguish. Might be more clear to include the examples from the manuscript in Supplementary Table 3 as well.

Answer: Thank you for your comments. We apologize for our misleading description. In our manuscript, our method can be applied to synthesize anilines from phenylboronic acids. If phenylboronic acid is replaced by pinacol ester (Bpin), we can also obtain aniline. Now, we revised the description. The related sentences are extracted as follows:

"A series of arylboronic acids with both electron-withdrawing and electron-donating substituents on the aryl ring are all amenable to our strategy, producing the corresponding amines in good yields. Aryl Bpins can also replace arylboronic acids to produce corresponding anilines with good yields under standard conditions, suggesting substrate applicability."

Comment 2: This sentence looks better broken up, however, it would read better if the sentences were shortened altogether, for example: "Coupling inorganic nitrogen sources (...) and carbon sources (...) to form upgraded organic amines is an emerging hot field in electrocatalytic C–N bond construction (refs)."

Answer: Thank you for pointing out this issue. We have revised the previous expression to a more appropriate form in the revised manuscript for better understanding. The revision is also extracted as follows: *"Coupling inorganic nitrogen sources (e.g., nitrogen (N₂), NO₃⁻, NO₂⁻) and carbon sources (e.g., carbon dioxide, carbon monoxide, and other carbonyl substrates) to form upgraded organic amines is an emerging hot field in electrocatalytic C–N bond construction"*.

Comment 3: Minor comment, but the part in the sentence “and most of them are small molecule reactions” seems redundant, that can be removed.

Answer: Thank you for your kind suggestion. We have removed the part in the sentence “and most of them are small molecule reactions” in the revised manuscript.

Comment 4: In the response, the authors mentioned the reactions for the counter electrode (HER and OER) however it still does not seem that this is mentioned in the manuscript or SI. This should either be depicted in the figure, or explained in text in one or both documents. Additionally, the bottom of Figure 2c is confusing, Cu(II) is consumed during cathodic time, so that part showing the Cu(II) being consumed should be on the bottom of figure 2b. As well, is the OH⁻ related to anodic time, or is the point of showing the OH⁻ just to show that the Ar-OH is not being formed? If it's the latter, I would make this more obvious. I think moving the “Cu(II) consumed” to cathodic time will help make this more obvious.

Answer: Thank you for pointing out this issue. We have added the relevant description in the general procedure of the revised manuscript as follows: “During the pulsed electrolysis process, the OER occurs at the cathodic time, and the HER occurs at the anodic time for the counter electrode”. In our work, pulsed electrochemistry causes the consumption of OH⁻ near the cathode surface because of the electrooxidation of Cu ($2\text{Cu} + 2\text{OH}^- - 2\text{e}^- \rightarrow \text{Cu}_2\text{O} + \text{H}_2\text{O}$, $\text{Cu}_2\text{O} + 2\text{OH}^- - 2\text{e}^- \rightarrow 2\text{CuO} + \text{H}_2\text{O}$, $\text{Cu} + 2\text{OH}^- - \text{e}^- \rightarrow \text{CuO} + \text{H}_2\text{O}$) at E_{an} , which can slow the change in pH, thus suppressing phenol byproducts. We have moved “Cu(II) consumed” to the bottom of Figure 2b in the revised manuscript.

Fig. R1 a) pulsed potential waveform, b) NO_2^- reduction before pulsed electrolysis, c) Cu(II) generation and Chan-Lam coupling under the anodic potential (E_{an}).

Comment 5: The sentence, as written, still doesn't make much sense, and the point being made here is still a little unclear. I agree with the logic in the sentence that has been added, however, I still think the first sentence doesn't seem to have much of a point. As it currently reads, it seems like the first sentence is a random statement of pH-related facts, and the sentence that was added in is explaining why MeOH leads to higher selectivity. It would make more sense to include a short introductory sentence explaining that you're looking at pH and solvent effects, along the lines of "The electrolyte solvent and pH play an important role in reaction selectivity."

Answer: Thank you for the kind suggestion. Based on the reviewers' wise suggestions, we have changed our description in the revised manuscript as follows:

"The electrolyte solvent and pH play an important role in reaction selectivity due to phenol byproducts being easy to form at a high concentration of OH^- ions in the Chan-Lam coupling reaction³²⁻³³. In addition, the efficient production of NH_3 from NO_2^- electroreduction is highly significant to the next C-N coupling step. NO_2^- electroreduction can proceed over the entire pH range, but the competing hydrogen evolution reaction (HER) lowers the ammonia selectivity and Faradaic efficiency (FE). The HER is serious in an acidic electrolyte, and phenylboronic acid is not stable in alkaline media⁴⁶⁻⁴⁷. Thus, the plused experiments are performed in a nearly neutral buffer. Furthermore, the hydrogen bonding interaction between MeOH and H_2O inhibits the H_2O dissociation process (the Volmer step) and slows the rate of migration of H_2O to the catalyst surface as hydrogen bonds⁴⁸, thus suppressing the HER and boosting the NH_3 FE from $\text{NO}_3^-/\text{NO}_2^-$ electroreduction. With this in mind, a mixture of MeOH and phosphate buffered saline (PBS) is adopted as the electrolyte for one-pot C-N coupling."

Comment 7: Mostly addressed, however the authors should indicate the potential that the potentiostatic conditions were run at.

Answer: Thank you for the kind suggestion. Electrolysis was carried out under potentiostatic conditions at -1.1 V vs. Hg/HgO for 7 hours to reduce nitrite to ammonia and then run under pulsed conditions. Detailed experimental steps have been added to the general procedure in the revised manuscript.

We highly appreciate the reviewer's thorough reading and constructive comments/suggestions about our manuscript!

REVIEWER COMMENTS

Reviewer #1 (Remarks to the Author):

Thank you for the manuscript revision and response to reviews questions. I have carefully read and evaluated the authors' comments, and my response are described below.

"However, the collection, transport, storage, and usage of NH₃ are time- and manpower-consuming processes that require high costs and complex handling. In addition, although the electrochemical reduction of NO₂⁻ to NH₃ has been widely reported, the electrosynthesis of organic amines from NH₃ generated in situ via NO₂⁻ electroreduction has rarely been studied owing to the lower reactivity over other nitrogen intermediates."

As I'm mentioning consistently throughout several rounds of revision, Chan-Lam coupling is not appropriate to make this point because Chan-Lam coupling cannot be performed on huge scale due to the high cost of arylboronic acid. For this sentence to make sense, reaction products should be in large demand and produced on enormous scale, for example, synthesis of urea or dehydrogenative coupling between benzene and ammonia.

"(1) Pulsed potentials enabled cascade conversion of NO₂⁻ and arylboronic acids to arylamines via NO₂⁻ electroreduction to NH₃ over a Cu cathode followed by in situ formed Cu(II)-catalyzed C-N coupling of NH₃ with arylboronic acids."

I believe the combination of electrochemical nitrite reduction and Chan-Lam coupling is new, but I'm not sure about importance of such new combination due to the reason discussed above.

"(2) The pulsed method avoids the use of molecular oxygen or organic oxidants."

Pulsed method is not the unique method to avoid the use of molecular oxygen. Ref 42 is a great example how simple electrochemical oxidation can be used to realise oxygen-free Chan-Lam.

"(3) Pulsed electrochemistry improves the selectivity of the products by adjusting the pH of the solution. Previous reports proved that phenylboronic acid was easy to hydrolyse into phenol byproducts in an aqueous solution (Tetrahedron Lett. 2003, 44, 1691-1694), and the phenol content increased with increasing pH, which was further confirmed by our experiment. In our work, pulsed electrochemistry causes the consumption of OH⁻ near the cathode surface because of the electrooxidation of Cu ($2\text{Cu} + 2\text{OH}^- - 2e^- \rightarrow \text{Cu}_2\text{O} + \text{H}_2\text{O}$, $\text{Cu}_2\text{O} + 2\text{OH}^- - 2e^- \rightarrow 2\text{CuO} + \text{H}_2\text{O}$, $\text{Cu} + 2\text{OH}^- - e^- \rightarrow \text{CuO} + \text{H}_2\text{O}$) at E_{an}, which can slow the change in pH, thus accelerating C-N formation and suppressing phenol byproducts."

Typical Chan-Lam coupling conditions do not produce so much phenol byproducts (otherwise synthetically not useful!). The conditions used in this work is strongly basic aqueous conditions, which seems to be responsible for the large amount of phenol formation. Such problem is definitely not

pronounced in standard Chan-Lam conditions as the water content is generally low and mild base is used. Therefore, reducing the amount of phenol byproducts by pulsed electrolysis seems to be addressing an issue that is peculiar to this system but generally not very problematic.

Accordingly, I'm still struggling to find strong novelty or practicality that justify the publication in Nature Communications.

Reviewer #2 (Remarks to the Author):

I believe that the authors have addressed all concerns. There is nothing more from this reviewer at this point.

Reviewer #3 (Remarks to the Author):

My comments have been addressed, this manuscript is ready for publication in Nature Communications.

A point-by-point response to the reviewers' comments

To reviewer 1:

Reviewer letter: As I'm mentioning consistently throughout several rounds of revision, Chan-Lam coupling is not appropriate to make this point because Chan-Lam coupling cannot be performed on huge scale due to the high cost of arylboronic acid. For this sentence to make sense, reaction products should be in large demand and produced on enormous scale, for example, synthesis of urea or dehydrogenative coupling between benzene and ammonia.

Comment 1: "Pulsed potentials enabled cascade conversion of NO_2^- and arylboronic acids to arylamines via NO_2^- electroreduction to NH_3 over a Cu cathode followed by in situ formed Cu(II)-catalyzed C-N coupling of NH_3 with arylboronic acids." I believe the combination of electrochemical nitrite reduction and Chan-Lam coupling is new, but I'm not sure about importance of such new combination due to the reason discussed above.

Answer: We appreciate the reviewer for recognizing the novelty of our work. We apologize that our description may have misled the core content of our manuscript. In addition, after considering the comments of the reviewer, we have provided a new reaction that can further highlight the utility of our work in the revised manuscript.

Our work focuses on developing a new construction method for electrocatalytic C-N bonds and exploring its mechanisms to achieve the green synthesis of organic amines from inorganic nitrogen sources (NO_2^-) under mild conditions and not a simple organic synthesis. Recently, electrochemical C-N bond construction from inorganic N sources (such as NO , NO_2^- or NO_3^-) has become a promising way to construct value-added organics and has become a research hotspot in catalytic and synthetic chemistry fields. (*Nat. Rev. Chem.* 2022, 6, 303-319; *Chem. Soc. Rev.* 2023, 52, 2193-2237; *Trends. Chem.* 2020, 2, 1004-1019). In the reactions reported thus far, the N species involved in C-N bond formation are active *N-containing intermediates produced during NO_2^- reduction, which are highly active and easily further reduced to NH_3 (*Nat. Sustain.* 2021, 4, 725-730; *J. Am. Chem. Soc.* 2022, 144, 16006-16011). Despite these achievements, the current studies are limited in reaction scope, and most of them are small molecule reactions, and the electrosynthesis of organic amines from NH_3 generated in situ via NO_2^- electroreduction has rarely been studied owing to the lower reactivity over other nitrogen intermediates, impeding methodology development and applications. In this regard, developing reactions to construct C-N bonds by utilizing NH_3 is significant to expand the C-N bond construction reaction scope and improve the synthesis efficiency for C-N bond products.

Chan-Lam amination provides an effective synthetic strategy for the formation of C-N bonds, and tremendous efforts have been devoted to improving the reaction since it was reported in 1998 (*Chem. Rev.* 2019, 119, 12491-12523; *Nat. Commun.* 2021, 12, 932; *Angew. Chem., Int. Ed.* 2015, 54, 6587-6590). Derek Lowe thought that "Chan-Lam is an appealing way of making C-N bonds, because it

is complementary to Pd-catalyzed processes like the Buchwald-Hartwig reaction” (The Chan-Lam Reaction, Tamed At Last. A chemical news published in Science). **It should be noted that price is not equal to value.** Despite the fact that phenylboronic acid as a necessary reactant is more expensive than the amine product, the Chan-Lam reaction is still under investigation, probably because of its mild reaction conditions, which can be carried out in air compared with the Pd-catalyzed Buchwald-Hartwig reaction. Additionally, the amine products from this reaction widely exist in pharmaceutical ingredients, agrochemicals, and natural products produced by the plant, **indicating excellent application value.** Based on the importance of the Chan-lam reaction, **it is necessary to develop the diversity and richness of synthesis methods.** Furthermore, the researchers at Eli Lilly used a continuous reaction setup to synthesize the desired product on a multikilo scale (over 50 kg) and further synthesize the active pharmaceutical ingredients, which is perhaps one of the best demonstrations of the use of Chan-Lam amination in the synthesis of bioactive molecules (*Org. Process Res. Dev.* 2019, 23, 1484–1498).

In this manuscript, we only selected the classical Chan-Lam coupling as a model reaction to demonstrate electrocatalytic C–N bond construction from inorganic nitrogen sources (NO_2^-) by the pulsed electrochemical method under mild conditions, avoiding the use of NH_3 and extra additional Cu(II) (*Angew. Chem., Int. Ed.* 2009, 48, 1114–1116). This not only represents an important advancement over classical Chan-Lam coupling but also broadens the application of the C–N bond construction reaction from inorganic nitrogen sources.

In addition, **we have provided a new reaction that can further highlight the utility of our work** in the revised manuscript *“Furthermore, the method can be applied to copper ion-catalyzed cycloaddition reactions for the one-pot two-step synthesis of 3,5-diphenyl-isoxazole by utilizing in situ generated benzaldoxime and phenylacetylene, namely, (i) the electroreduction of NO_2^- to NH_2OH , followed by benzaldehyde to benzaldoxime over an LC-Cu NC cathode, and (ii) subsequent synthesis of 3,5-diphenyl-isoxazole from benzaldoxime with phenylacetylene catalyzed by in situ generated Cu(II) (Supplementary Fig 20). In addition, the pulsed electrochemical protocol can be expanded to the click reactions of terminal alkynes with benzyl azides to construct 1,2,3-triazole N-heterocycles, **implying good methodology universality**”.*

Fig. R1 One-pot synthesis of 3,5-diphenyl-isoxazole from benzaldehyde with NO_2^- and phenylacetylene by a pulsed electrochemical method.

Comment 2: "The pulsed method avoids the use of molecular oxygen or organic oxidants."

Pulsed method is not the unique method to avoid the use of molecular oxygen. Ref 42 is a great example how simple electrochemical oxidation can be used to realise oxygen-free Chan-Lam.

Answer: Thank you very much for the reviewer's comments. Electrochemical oxidation can indeed be used to achieve oxygen-free Chan-Lam. To further verify the benefits of the pulsed step in our work, the potential was switched to 0.4 V vs. Hg/HgO after NO_2^- electroreduction to NH_3 for Chan-Lam coupling. As shown in Fig. R2a, the yield of aniline is only 16%, less than that obtained by pulse electrolysis, which may be due to NH_3 being more easily oxidized than aniline, which is the reactant in Ref 42 (Supplementary Fig. 10), and the yield of NH_3 decreases gradually with increasing anodic potential (Supplementary Fig. 11), resulting in a lower yield of aniline. Pulsed electrochemical catalysis has been demonstrated as an operable and efficient reaction regulation method in complicated electrocatalytic reactions due to its precise control of electron transmission and regulation selectivity in recent years (*Angew. Chem. Int. Ed.* 2023, 62, e202217635). In our work, the pulsed electrochemical method can form Cu(II) in situ and maintain high concentrations of Cu(II) without molecular oxygen or other chemical oxidants at anodic potential and form NH_3 in situ and maintain enough concentrations of NH_3 for the Chan-lam reaction at the cathodic potential. The above results prove that the pulsed method can facilitate the construction of the C–N bond.

Fig. R1 **a** Comparison experiment using the electrochemical oxidation method. **b** LSV curves of LC-Cu NCs at a scan rate of 5 mV s⁻¹ with aniline **2a** and ammonia. **c** Yield of NH₃ under pulsed electrolysis conditions with E_{ca} = -1.1 V, different E_{an} values, and t_{an} = t_{ca} = 2 s.

Comment 3: "Pulsed electrochemistry improves the selectivity of the products by adjusting the pH of the solution. Previous reports proved that phenylboronic acid was easy to hydrolyse into phenol byproducts in an aqueous solution (*Tetrahedron Lett.* 2003, 44, 1691–1694), and the phenol content increased with increasing pH, which was further confirmed by our experiment. In our work, pulsed electrochemistry causes the consumption of OH⁻ near the cathode surface because of the electrooxidation of Cu (2Cu + 2OH⁻ - 2e⁻ → Cu₂O + H₂O, Cu₂O + 2OH⁻ - 2e⁻ → 2CuO + H₂O, Cu + 2OH⁻ - e⁻ → CuO + H₂O) at E_{an}, which can slow the change in pH, thus accelerating C–N formation and suppressing phenol byproducts." Typical Chan-Lam coupling conditions do not produce so much phenol byproducts (otherwise synthetically not useful!). The conditions used in this work is strongly basic aqueous conditions, which seems to be responsible for the large amount of phenol formation. Such problem is definitely not pronounced in standard Chan-Lam conditions as the water content is generally low and mild base is used. Therefore, reducing the amount of phenol byproducts by pulsed electrolysis seems to be addressing an issue that is peculiar to this system but generally not very problematic.

Answer: Thank you for the reviewers' comment. Although the conventional Chan-Lam synthesis method does not produce many phenol byproducts, most works are performed in organic solvents, which are costly and less environmentally benign (*Org. Lett.* 2012, 14, 4326–4329; *J. Am. Chem. Soc.* 2017, 139, 4769–4779; *Chem. Rev.* 2019, 119, 12491–12523). In addition, some reactions have required heating to 40–120 °C (*Chem. Commun.* 2013, 49, 8359–8361; *Tetrahedron Lett.* 2015, 46, 4843–4847). The Chan-Lam reaction involves the coupling of NH₃ with arylboronic acids to form aniline (*Angew. Chem. Int.*

Ed. 2009, 48, 1114–1116), and $\text{NH}_3 \cdot \text{H}_2\text{O}$ is a weak alkaline solution, which inevitably causes the hydrolysis of phenylboronic acid. Furthermore, the preparation of NH_3 that dominantly relies on the Haber-Bosch process is highly energy intensive with a large amount of CO_2 emissions (*Energy Environ. Sci.* 2016, 9, 2550–2554; *Nature* 2019, 570, 504–508; *Nat. Chem.* 2017, 9, 64–70). Additionally, the collection, transport, storage, and usage of NH_3 are time- and manpower-consuming processes. Therefore, searching for an efficient, inexpensive and sustainable method for Chan-Lam at room temperature (RT) is highly significant.

Electrochemical C–N coupling by integrating $\text{NO}_3^-/\text{NO}_2^-$ electroreduction to NH_3 in aqueous media is an attractive approach due to the availability of water as the hydrogen source, renewable electricity, and environmentally benign reaction conditions. However, OH^- is inevitably generated when conducting the NO_2^- electroreduction ($\text{NO}_2^- + 5\text{H}_2\text{O} + 6\text{e}^- \rightarrow \text{NH}_3 + 7\text{OH}^-$), which causes the hydrolysis of phenylboronic acid. Therefore, in our work, the pulsed anodic potential was used to slow the change in pH by the consumption of OH^- through OER ($4\text{OH}^- \rightarrow 2\text{H}_2\text{O} + \text{O}_2 + 4\text{e}^-$) or the electrooxidation of Cu ($2\text{Cu} + 2\text{OH}^- - 2\text{e}^- \rightarrow \text{Cu}_2\text{O} + \text{H}_2\text{O}$, $\text{Cu}_2\text{O} + 2\text{OH}^- - 2\text{e}^- \rightarrow 2\text{CuO} + \text{H}_2\text{O}$, $\text{Cu} + 2\text{OH}^- - 2\text{e}^- \rightarrow \text{CuO} + \text{H}_2\text{O}$) and inhibit phenol byproducts, resulting in the efficient synthesis of aniline with high yield.

In summary, although alkaline aqueous solutions produce phenol byproducts, the pulsed method is used to slow down the pH change to suppress phenol byproducts and accelerate the construction of C–N bonds, thus achieving efficient conversion from an inorganic nitrogen source (NO_2^-) to organic amines with high yield under mild conditions. The same effect as other conventional methods is achieved and avoids the use of additional NH_3 , copper ions, and molecular oxygen, which is economical and green.

Department of Chemistry
Tianjin University
Tianjin 300072, P. R. China
Tel&Fax: 86-22-27403475
E-mail: bzhang@tju.edu.cn

To reviewer 2:

Reviewer letter: I believe that the authors have addressed all concerns. There is nothing more from this reviewer at this point.

Answer: We highly appreciate the reviewer for his/her positive comments on our revised manuscript. We are sure that the quality of this work has been greatly improved according to these comments and suggestions.

Department of Chemistry
Tianjin University
Tianjin 300072, P. R. China
Tel&Fax: 86-22-27403475
E-mail: bzhang@tju.edu.cn

To reviewer 3:

Reviewer letter: My comments have been addressed, this manuscript is ready for publication in Nature Communications.

Answer: We highly appreciate the reviewer for his/her positive comments on our revised manuscript. We are sure that the quality of this work has been greatly improved according to these comments and suggestions.